# A Comprehensive Study from Cradle-to-Grave on the Environmental Profile of Malted Legumes

**DOI:** 10.3390/foods13050655

**Published:** 2024-02-21

**Authors:** Mauro Moresi, Alessio Cimini

**Affiliations:** Department for Innovation in the Biological, Agrofood and Forestry Systems, University of Tuscia, Via S. C. de Lellis, 01100 Viterbo, Italy; a.cimini@unitus.it

**Keywords:** dry pulses as such or malted and decorticated, environmental profile, life cycle analysis, mitigation actions, overall weighted sustainability score, PEF standard method

## Abstract

Three representative pulses from the Latium region of Italy (namely, *Solco Dritto* chickpeas, SDC, *Gradoli Purgatory* beans, GPB, and *Onano* lentils, OL) underwent malting to reduce their anti-nutrient content, such as phytic acid and flatulence-inducing oligosaccharides. This initiative targets the current low per capita consumption of pulses. Employing Life Cycle Analysis, their environmental impact was assessed, revealing an overall carbon footprint of 2.8 or 3.0 kg CO_2e_ per kg of malted (M) and decorticated (D) SDCs or GPBs and OLs, respectively. The Overall Weighted Sustainability scores (OWSS) complying with the Product Environmental Footprint method ranged from 298 ± 30 to 410 ± 40 or 731 ± 113 µPt/kg for malted and decorticated SDCs, OLs, or GPBs, indicating an increase from 13% to 17% compared to untreated dry seeds. Land use impact (LU) was a dominant factor, contributing 31% or 42% to the OWSS for MDSDCs or MDOLs, respectively. In MDGPBs, LU constituted 18% of the OWSS, but it was overshadowed by the impact of water use arising from bean irrigation, accounting for approximately 52% of the OWSS. This underscores the agricultural phase’s pivotal role in evaluating environmental impact. The climate change impact category (CC) was the second-largest contributor, ranging from 28% (MDSDCs) to 22% (MDOLs), and ranking as the third contributor with 12% of the OWSS for MDGPBs. Mitigation should prioritize the primary impact from the agricultural phase, emphasizing land and water utilization. Selecting drought-tolerant bean varieties could significantly reduce OWSSs. To mitigate climate change impact, actions include optimizing electricity consumption during malting, transitioning to photovoltaic electricity, upgrading transport vehicles, and optimizing pulse cooking with energy-efficient appliances. These efforts, aligning with sustainability goals, may encourage the use of malted and decorticated pulses in gluten-free, low fat, α-oligosaccharide, and phytate-specific food products for celiac, diabetic, and hyperlipidemic patients. Overall, this comprehensive approach addresses environmental concerns, supports sustainable practices, and fosters innovation in pulse utilization for improved dietary choices.

## 1. Introduction

Legumes, known for their abundant protein, dietary fiber, and micronutrient content, are being increasingly utilized in the preparation of pulse-based food products as substitutes for animal-derived foods or as ingredients for gluten-free alternatives [1]. Despite their remarkable nutritional content [2] and environmentally friendly cultivation practices [3], the global per capita legume consumption has remained stagnant for the past three decades, hovering at a mere 21 g per day [4]. Several factors contribute to this limited uptake, including lengthy cooking times, an unappealing taste, proteins with low digestibility, gastrointestinal issues [5], and a notable presence of anti-nutrients, such as phytic acid, tannins, enzyme inhibitors, and oligosaccharides that induce flatulence [6,7,8].

A recent online survey explored how consumers in Denmark, Germany, Poland, Spain, and the United Kingdom perceive and use various types of pulses [9]. The study found that lentils, kidney beans, and chickpeas are the most popular pulses, commonly consumed at home in dried or canned form. Despite the general belief among participants that pulses are healthy and natural, a significant majority remained uninformed about the low environmental impact linked to pulse production and consumption. This underscores the necessity for effective promotional strategies featuring clear communication to enlighten consumers and rectify this prevailing misconception [9].

Legumes, through symbiosis with nitrogen-fixing bacteria, diminish reliance on synthetic N fertilizers [10], thereby lowering greenhouse gas emissions in agriculture [11]. Consequently, pulse crops assume a pivotal role as break crops in cereal-dominated rotations, improving soil structure and yields in subsequent cereal crops [12]. Additionally, they mitigate issues such as weeds, pests, and diseases [13], while also contributing to the reduction in energy consumption, global warming potential, ozone formation, acidification, and environmental and human toxicity [3]. Furthermore, pulses demonstrate favorable nitrogen investment factors, with approximately 1.2 kg of new reactive nitrogen needed to produce one unit of nitrogen in pulse grains. It is worth noting that this factor is notably lower in sugar beet, fruits, vegetables, and potatoes (around 2 g N per kg N), while it peaks at 15–20 kg of N per kg of N in beef [14]. Overall, the practice of rotating legumes in crops presents intriguing possibilities for alleviating environmental pressures, particularly in the context of diminishing fossil energy resources and the challenges posed by climate change [3].

To reduce the anti-nutrient levels in pulses [15], including lentils [16], a variety of traditional techniques (such as milling, dehulling, soaking, boiling, pressure cooking, sprouting, and fermentation) and emerging methods (such as dielectric heating, extrusion, *γ*-irradiation, ultrasound, and high hydrostatic pressure) have been employed, resulting in varying degrees of reduction. Among these, the malting process, commonly used in beer production, activates several hydrolytic enzymes that decrease key anti-nutrients, such as phytic acid responsible for mineral malabsorption and oligosaccharides causing flatulence, in certain pulses [17,18].

The main objective of this study was to conduct a business-to-consumer (B2C) life cycle assessment (LCA) for three typical pulses cultivated in the Latium region. These pulses are locally distributed in dry form and may potentially be commercialized in malted form as functional foods with improved nutritional characteristics and non-flatulence-inducing properties.

Various standardized approaches exist to evaluate the environmental footprint of food items, as discussed by Moresi et al. [19]. For example, the Publicly Available Specification (PAS) 2050 method [20] calculates the greenhouse gas emissions associated with a specific product, known as its carbon footprint. Other standard methods cover a range of environmental impact categories, including acidification, eutrophication, stratospheric ozone depletion, and photochemical ozone creation, among others.

In this study, the Product Environmental Footprint standard method was employed, as it provides an end-point-oriented approach to measuring the cradle-to-grave environmental performance of goods or services throughout their life cycle. This method originated from the European Commission’s proposal to “establish a common methodological approach to enable Member States and the private sector to assess, display, and benchmark the environmental performance of products, services, and companies based on a comprehensive assessment of environmental impacts over the life cycle” [21].

Thus, the primary aim of this study was to compare, for the first time, the environmental profiles of novel low-anti-nutrient-containing pulses and conventional dry ones, identify the main hotspots in their life cycle, and propose potentially effective mitigation actions.

## 2. Pulses: Cultivation and Utilization, Market Prospects, and Ecological Implications

In 2022, global pulse production reached nearly 96 million metric tons (Mg), with dry beans, chickpeas, and lentils contributing approximately 27.7 million Mg, 14.3 million Mg, and 5.7 million Mg, respectively [22]. India leads in dry bean and chickpea production, yielding around 6.1 million Mg and 9.9 million Mg annually, respectively. Canada ranks as the primary lentil producer, with an estimated annual production of 3.23 million Mg. Turkey follows as the second-largest producer of dry beans (630,000 Mg), Brazil of chickpeas (2.9 million Mg), and India of lentils (1.06 million Mg).

In 1960, Italian dried legume production peaked at 640,000 Mg but fell to 135,000 Mg in 2010 [23]. Despite increased production, Italy relies heavily on imports, constituting about 95% of bean consumption, 59% of chickpea consumption, and 98% of lentil consumption. In 2023, the Italian annual production of dry beans, chickpeas, and lentils was approximately 10,665 Mg, 24,036 Mg, and 4565 Mg, respectively (http://dati.istat.it/Index.aspx?QueryId=37850, accessed 11 February 2024).

Amidst the ongoing efforts to document, conserve, and share the genetic heritage of Italy’s Latium region, with a concurrent focus on protecting the biodiversity of animal and plant species possessing distinct and irreplaceable traits [24], there is a particular interest in valorizing three indigenous pulses: *Gradoli Purgatory* beans (GPB), *Solco Dritto* chickpeas (SDC), and *Onano* lentils (OL). GPBs resemble Cannellini beans but with a thinner skin, SDCs are smooth, yellow-beige-skinned seeds, and OLs are round and light brown with marbled surfaces. Traditionally grown in Viterbo province, particularly in Gradoli, Acquapendente, and Onano, these pulses thrive in hilly volcanic soils at 300–400 m above sea level, benefiting from the mild climate near Lake Bolsena. SDCs and OLs trace their origins to the Etruscans, while GPBs became popular due to the traditional Purgatory lunch served in Gradoli every Ash Wednesday since the 17th century [25]. SDCs are named after the furrow tracing beneath the town of Valentano on August 14th, signaling a bountiful harvest [26]. OLs earned a Protected Geographical Indication (PGI-IT-02651) status from the European Commission in 2022 [27].

In previous studies [17,18,28], the primary operational factors for the three stages—soaking, germination, and kilning—of the malting process for such dry pulse varieties were determined at the laboratory scale.

Table 1 displays the dry basis composition of SDC, GPB, and OL seeds, in both their original and malted forms, as extracted from references [17,18].

Additionally, such a process not only yielded a greater availability of absorbable molecules from the indigestible proteins in legumes, as revealed by the increased levels of free amino acids in malted pulses [29], but also enhanced the cooking and nutritional characteristics of malted chickpeas [30] and beans [31]. Additionally, dehulled malted pulse flour was utilized to create fresh egg pasta with a high raw protein content (20–24 g/100 g), low phytate levels (0.6–0.8 g/100 g), and an in vitro glycemic index (GI) of 28–41%, devoid of flatulence-inducing oligosaccharides [18,32]. However, the fresh egg pasta including malted GPB flour exhibited not only a significantly smaller glycemic index (28% ± 3%), but also a resistant starch–total starch ratio far greater than the threshold value (14%) specified by the European Commission Regulation 432/2012 [33] to label foods with a health claim indicating improvement in postprandial glucose metabolism.

Currently, there are no studies evaluating the environmental impact of such niche pulse production. It is well known that the environmental impact of cultivating pulses is influenced by factors such as climate conditions, species, variety, and the production system (organic or conventional). Additionally, the processing, packaging, and transportation of pulses contribute significantly to their overall environmental footprint. Recognizing the crucial role of pulses in sustainable food production, numerous studies have employed the Life Cycle Assessment (LCA) methodology to evaluate and communicate their environmental performance. Some investigations have focused solely on the cradle-to-gate field phase, neglecting the consumer use phase [34,35,36]. In contrast, others have undertaken a comprehensive approach, analyzing the entire pulse supply chain from cradle to distribution center [37] or even extending to end of life, encompassing the use and post-consumer waste disposal phases [38,39]. Certain studies have specifically accounted for greenhouse gas emissions only [9,40,41], while others have considered a broader spectrum of impact categories [35,37,38,39].

In their evaluation of the environmental impact of selected pulses (such as beans, chickpeas, and peas) when packaged in glass bottles or steel tin cans (both as singles and in multipacks, including cardboard cluster packs), del Borghi et al. [37] identified the production of packaging as the primary hotspot, contributing to over 70% of overall impacts. This high percentage was attributed to the substantial non-renewable energy consumption associated with the manufacture of packaging materials, specifically glass and steel. Pulse processing accounted for 13% of the total impacts, primarily due to the use of natural gas during the sterilization cooking step. In a comprehensive cradle-to-grave assessment of various pulses, considering both dry and canned forms, Tidåker et al. [39] found that energy use related to cultivation made a relatively minor contribution (8–36%) to overall energy consumption. This was particularly evident when considering cooking, packaging, and transportation. The transportation phase exerted a significant influence on energy use, especially for imported pulses, notably those processed in Italy and transported in canned form to Sweden. For canned pulses, the energy use associated with retorting was almost negligible compared to the energy expended in the production and subsequent management of packaging waste. In contrast, for dry pulses, the energy consumption linked to home cooking was 3–6 times higher than that for producing the packaging. Bandekar et al. [38] conducted a cradle-to-grave assessment of the environmental impact associated with the production and consumption of pulses (e.g., field pea, lentil, chickpea, and dry bean) in the USA. The study emphasized the considerable influence of the consumption phase whatever the cooking method (i.e., open vessel and stovetop pressure cooker, each one heated by an electric range, and electric pressure cooker) used at the consumer stage. The study revealed that cooking time and energy use efficiency play crucial roles in determining electricity consumption during this phase. Notably, longer cooking times and a lower energy use efficiency resulted in a higher environmental impact, while shorter cooking times and a higher energy use efficiency led to a more sustainable outcome. In practical terms, adopting practices such as using electric pressure cookers or preparing larger batches of legumes might significantly reduce this environmental impact, promoting a more sustainable approach to legume consumption.

Thus, this LCA study will be the first one dealing with the environmental profiles of novel low-anti-nutrient-containing pulses as compared to those of conventional dry ones.

## 3. Methodology

This study adhered to the Life Cycle Assessment procedure outlined in the ISO standards 14040 [42] and 14044 [43], encompassing the following stages: goal and scope definition, inventory analysis, impact assessment, and interpretation of results.

### 3.1. Goal and Scope Definition

The objective of this study was to create a Life Cycle Assessment model for evaluating the mid- and end-point environmental profiles of three malted and decorticated pulses from the Latium region of Italy. In this way, the profiles of these innovative low-anti-nutrient-containing pulses were compared to those of conventional dried pulses, thus identifying the main hotspots in their life cycle. The selected functional unit (FU) was a modified-atmosphere polypropylene (PP) bag containing 500 g of legumes, available in either dried or malted and decorticated form, commonly sold in supermarkets and retailers in Italy’s Latium region.

In Figure 1, the system boundary diagram depicts the life cycle from cradle to grave for dry legumes in their natural state. The main processes were subdivided into upstream, core, and downstream ones and included the following steps:

(i)Conventional cultivation of legumes in the Latium region of Italy.(ii)Pre-treatment of the harvested seeds to prevent insect proliferation.(iii)Cleaning and selection of treated legumes, which are then stored in polyethylene (PE) super-sacks.(iv)Primary packaging of selected legumes in food-grade PP bags, where inside air is automatically removed and replaced with gaseous N_2_. These are then sealed with a cardboard collar and two brass rivets to extend the shelf life of pulses to at least 1 year.(v)Secondary and tertiary packaging: twelve 500 g legume packages are collected in each cardboard box, which is then palletized.(vi)Storage at room temperature.(vii)Transportation of palletized products using a Jumper-type van for delivery to retailers, where the primary packages are displayed on store shelves.(viii)Home consumption: the packaged lentils are directly cooked in 4 L of tap water for each kg of dry legume, which is kept boiling for the time recommended on the label. In contrast, dried beans and chickpeas are similarly cooked on the condition that they have been preliminarily soaked in tap water (4 L/kg) for 16 h.(ix)Disposal of post-consumer cooked legume and packaging wastes.(x)Figure 1 also shows a block diagram related to the production of malted and decorticated legumes from cleaned and selected legumes (step iii), this involving the following steps:(xi)Rehydration of dry legumes in water for 2 h in the case of Onano lentils [18] or 3 h in the case of Gradoli Purgatory beans or Valentano straight furrow chickpeas [17].(xii)Germination under pre-set thermo-hygrometric conditions (25 °C) for 72 h to reduce the phytic acid and oligosaccharide contents.(xiii)Drying (kilning) of germinated legumes with a pre-established thermal diagram.(xiv)Decortication of malted legumes using a cyclone separator to remove their cuticles and rootlets.(xv)Optical selection of malted pulse cotyledons.

The subsequent processing steps coincide with those described for dried legumes, namely packaging (steps iv–v), room storage (step vi), distribution (step vii), consumption (step viii), and post-consumption (step ix).

The production of capital goods (machinery, etc.), as well as their cleaning and disposal, any personnel travel, and the transport of consumers to and from points of purchase were excluded from the system boundary, as suggested by Sections 6.5 and 6.4.4 of PAS 2050 standard method [20]. Moreover, the LCA was referred to the year 2022, while the process technology underlying the datasets used in this study reflected the process configurations typical for dry pulse processing on an industrial scale in the reference year.

The primary data for pulse cultivation and processing were provided by Il Cerqueto Srl (Acquapendente, Italy), whereas the secondary data were extracted from the Ecoinvent v. 3.9.1 database using the allocation, cut-off, system model [44], which was incorporated into the LCA software Simapro 9.5.0.0 (Prè Consultants, Amersfoort, The Netherlands) and other technical reports, as detailed below. Both primary and secondary data were employed to identify six distinct product stages (processes, assembly, reuse waste scenarios, end-of-life scenarios, and life cycles) for constructing the dried pulse network using SimaPro 9.5.0.0 software. Additionally, the end-of-life scenarios of primary, secondary, and tertiary packaging materials, along with the tertiary package (semi-pallet), were integrated into parallel life cycles. A summary of these product stages is provided in the electronic Appendix A and will be elaborated further below.

### 3.2. Life Cycle Inventory Analysis

#### 3.2.1. Pulse Cultivation

A summary of the cultivation conditions used to grow the three varieties of legumes at the Cerqueto Agricultural Farm (Acquapendente, Italy) is presented in Table 2.

Appendix A in the electronic version of Appendix A shows the input and output data regarding the conventional production of dry pulses at the aforementioned agricultural farm, as referred to a nominal area of 1 ha, together with the GHG emissions from managed soils, as estimated according to the recently updated IPCC Guidelines [45] using the TIER I approach. Specifically, the emissions of NO into air and P into water were estimated using the emission factors reported in the Product Category Rules EPD^®^ [46], as indicated in Appendix A. The NPK fertilizer used was assumed to be made of ammonium nitrate and calcium ammonium nitrate in equal percentage shares of 50%, while poultry manure was assimilated into an animal manure. Additionally, P emissions were estimated as 5% of the applied P with mineral and organic fertilizers.

The inventory associated with SDC cultivation is exemplified in Appendix A.

#### 3.2.2. Grain Fumigation

After harvest, legumes are directly shelled in the field, resulting in a substantial volume of agricultural waste comprising empty pods, leaves, and stems. The seeds transported to the processing facility for industrial handling did not undergo additional drying, as their average moisture content was around 12% (*w*/*w*). Legumes from conventional agriculture undergo fumigation with phosphine gas (PH_3_) in evacuated and sealed environments, maintaining temperatures between 10 and 30 °C and employing gas detection sensors. Compressed tablets, each weighing 3 g and composed of 56% (*w*/*w*) aluminum phosphide (AlP) along with various formulations (ammonium carbamate, ammonium bicarbonate, urea, and paraffin), are utilized to control fumigant release and mitigate flammability. In the presence of moisture, aluminum phosphide liberates phosphine and aluminum hydroxide:AlP + 3 H_2_O → PH_3_↑ + Al(OH)_3_(1)

Thus, about 1 g of PH_3_ was released from each tablet. Typically, from 3 to 6 tablets per metric ton (Mg) of pulse grains are applied when piled and covered with plastic sheets. Following phosphine release, the residue primarily consists of aluminum hydroxide, with slight amounts of undecomposed aluminum phosphide potentially remaining in the white-gray powder residue from the tablets, pellets, or sachets [47,48]. At the Cerqueto factory (Acquapendente, Italy), 100 g of phosphine is utilized for 20 Mg seed batches (equivalent to 5 g of phosphine/Mg of legume seeds), translating to 5 tablets per Mg of grains. The fumigation process typically spans 7 days. Conversely, organic legumes undergo a −25 °C infestation treatment lasting 30 days.

The inventories associated with the production of AlP tablets and fumigated pulse grains are shown in Appendix A, respectively.

#### 3.2.3. Seed Cleaning/Grading

The qualitative selection of pulse grains involves a preliminary density-based separation to isolate mature grains, regardless of size, as legume density typically increases with maturation [49]. Additional residues are obtained during size-based separation and in optical selection stages, as well as during the final manual selection before the preservation process [50]. Generally, by-products generated during this phase include discarded seeds due to small size, defective seeds based on color or breakage, and cuticles. Table 3 provides the minimum, maximum, and average percentages of cleaning waste for the three legumes under consideration, along with the average fractions of dust, grass and insects, and broken grains, etc., and the corresponding yields of cleaned, ready-to-be packed grains per hectare [51]. The cleaning residues are pulverized, separated from the herbaceous and insect-rich fraction, and ultimately pelletized for use in animal husbandry.

Although there is an interest, in line with circular economy concepts, in utilizing all by-products (seed hulls, pods, broken seeds, protruding roots, etc.) from legume processing as ingredients or raw materials for extracting bioactive compounds [52], in this study, it was assumed that they were disposed of as animal feed, as small producers of dry legumes are deemed unsuitable for any profitable valorization. The recovered powders (PO) and herbaceous fraction (E) are instead returned to agricultural soil. The so-selected legumes are initially stored in polyethylene (PE) super-sacks, each weighing about 3 kg with a load capacity of 1 Mg. These super-sacks are a type of flexible intermediate bulk container (FIBC) that can be loaded and unloaded from any angle in times as short as from 50 to 75% of the typical ones. They are stored at room temperature in the company warehouse, awaiting the packaging phase for dry legumes. The inventory associated with the production of such super-sacks is shown in Appendix A.

Once emptied, these super-sacks cannot be reused for storing consumable products but are repurposed to collect pelletized cleaning residues (SSU), consisting of RP and pulse waste (SGP) formed during primary packaging (Figure 2), to be sent to livestock farms. The end-of-life of PE super-sacks involves their disposal as plastic waste. During this intermediate storage, the cleaned dry legumes experience an average weight loss of 4%, resulting in a reduction in the average moisture content of cleaned legumes from 12% (*w*/*w*) to 8.3% (*w*/*w*). In Figure 2, a block diagram of the fumigation process, cleaning and optical selection of harvested legumes, and storage of cleaned grains is presented. All symbols used to identify the various stream flows are given in the Nomenclature section. Appendix A provides the material balance related to the average yield of the harvested fresh grain pulses shown in Table 2.

As the emissions associated with legume cultivation are generally allocated on an economic basis, Appendix A presents the current selling prices of the cleaned and selected grains (GPD) and by-products (MZ) designated for animal husbandry. It is noteworthy that the emissions are practically associated solely with legume production, with an allocation level exceeding 99.6%. Appendix A shows the inventory associated with the production steps of FIBC-packed cleaned dry pulse grains, while Appendix A displays the inventory associated with the production of ready-to-pack dried pulses.

#### 3.2.4. Packaging of Dry Pulses

Dry legumes are conveyed to the packaging machine, where they are packed into food-grade polypropylene (PP) bags, creating a modified atmosphere inside. The machine removes the existing air and, before sealing, introduces N_2_ into the package, which is then closed with a cardboard collar and two brass rivets and, finally, labeled. This type of packaging extends the shelf life of dry legumes at room temperature for up to one year. To this end, compressed nitrogen gas in rechargeable cylinders (with an outer diameter of 203 mm, a height of 1650 mm, and a gross weight, including the valve and cap, of ~54 kg, containing about 10 kg of nitrogen gas at a pressure of 200 bar: https://www.tecnoproject.com/documents/30774/82464/azoto_modalit%C3%A0.pdf/cb4932ed-756a-09a3-54d7-445912d232ae; access on 12 February 2024) is utilized. On average, 4 cylinders of 40 L each are consumed to package 4 Mg of dry legumes. Therefore, specific consumption, including losses during the packaging machine operation, is around 0.01 kg of N_2_ per kg of dry legumes.

Each paper label, taken from a reel, weighed 0.808 g, 40% of which represents the mass of the effective adhesive label (0.323 g) and the remaining 60% the supporting cardboard (0.485 g), which was disposed of as paper and cardboard waste. Twelve bags were collected in a carton of recycled cardboard (CA), which was closed with scotch tape and labelled. The tertiary packaging comprised a semi-pallet made of anthracite-colored recycled polypropylene, over which different layers of cartons were stacked, tightened with 3 wraps of stretch-and-shrink PE film and labeled with two adhesive paper tags. Figure 3 shows a block diagram for the packaging process examined in this work, also showing all the solid wastes generated, while Table 4 gives all the details about the primary, secondary, and tertiary packages used. Appendix A presents the average waste of cleaned-ready-to-pack pulses and packaging materials recorded during the factory production on a year basis.

The material balance of the packaging process of dried legumes and their waste management is summarized in Appendix A. The inventory associated with all packaging items is shown in Appendix A. The assemblies of primary, secondary, and tertiary packages are detailed in Appendix A, while those of primary and secondary packaging and primary, secondary, and tertiary packaging in Appendix A. Finally, the assembly of dry pulses is described in Appendix A.

#### 3.2.5. Logistics of Input and Output Materials

Table 5 shows the logistics of the input/output materials with the type and load of the means of transport used and overall distance travelled from the places of production to those of use/delivery, as mainly derived from the dry pulse processing plant of reference.

#### 3.2.6. Energy Sources

Electricity is the sole energy resource used to produce dried legumes, as it is used to operate equipment for cleaning and optically select the grains and packaging machinery, as well as for lighting, heating, and conditioning the premises of the production plant. It was drawn from the medium-voltage Italian grid. In 2022, the reference factory (Il Cerqueto Srl, Acquapendente, Italy) consumed around 32,900 kWh to process ~50 Mg of dried pulses, this being equivalent to about 0.66 kWh of electricity per kg of dried pulses packed.

#### 3.2.7. Consumer Use

Dried legumes in nitrogen-sealed bags are generally stored at room temperature for at least one year. Their cooking process can be divided into two distinct phases. The first phase involves soaking the dried legumes in tap water for 16 or 24 h [53], while the second one involves cooking the legumes in boiling water using a water-to-dried legume ratio of 4 L/kg [54]. In reality, the latter can be distinguished into two subphases. The first one involves heating the mass of water and dried legumes from room temperature to boiling point, while the second one is carried out at such a temperature to make the legumes easily chewable. Such a cooking time is a function of the seed size and type and can be assessed based on the cooked legume hardness. This can be determined by chewing by a panel of trained tasters, by pinching between the thumb and index fingers of tasters according to the so-called pinch test [55], or by compression and extrusion through a mini-Kramer shear cell [53] or an Ottawa Texture Measuring System [30], loaded with 7.50 ± 0.5 g or 70.0 ± 0.5 g of cooked legumes, respectively, using a texture analyzer.

Table 6 summarizes the consumption modes for the three legumes examined here, either in their commercial form or malted and decorticated form, together with their soaking and cooking times and specific energy consumption (e_C_). Further details about the assessment of the effective cooking times of pulses and the overall cooking energy consumption are reported in Appendix B by referring to several papers [30,31,54,56,57,58,59].

In the European Union, 83% of household kitchens use gas stoves, while the remaining 17% use electric stoves [60]. Thus, the specific cooking energy of legumes is proportionally distributed between such gas and electric stoves in use on the assumption of withdrawing natural gas or electricity from the national network or low-voltage grid, respectively.

Due to their protein and moisture content, both cooked legumes and dishes containing cooked legumes should not be kept at room temperature for more than 2 h. They might be stored in the refrigerator at around 4 °C for no more than 3 days, provided that they are reheated at the innermost point to 73 °C for a few minutes before serving.

For the purpose of this study, a standard serving of cooked legumes, equivalent to 50 g of dry legumes [61], was assumed to be served in a 400 g ceramic bowl. The production of these bowls was modeled based on the process used for sanitary ceramic available in the Ecoinvent database v. 3.9.1, following the approach suggested by Martin et al. [62]. It was assumed that these bowls were manufactured in Tuscany, with raw materials transported from the extraction site to the factory gate using EURO5 medium-duty trucks, covering an average distance of about 100 km. Subsequently, the final product was conveyed from the factory gate to the points of sales using similar trucks, spanning an average distance of approximately 300 km. In terms of usage, approximately 2000 usage cycles were considered for each bowl, following the recommendation by Rahat [63]. Additionally, bowl washing was conducted using a dishwasher with a C energy class rating (https://www.candy-home.com/it_IT/lavastoviglie/32002327/cf-4c6f0w/; accessed 12 February 2024), featuring a load capacity of 14 place settings and specific consumptions of electricity, tap water, and detergent at 0.74 kWh, 10.9 L, and 10 g per cycle, respectively. Typically, a dishwasher cycle includes 2 plates, 1 saucer, 1 glass, 1 coffee cup with a saucer, and 5 pieces of cutlery. In this specific case, which involved the use of a bowl, a glass, and a spoon, the load capacity of the dishwasher was assumed to be approximately equivalent to 28 place settings per cycle.

#### 3.2.8. Disposal of Processing and Post-Consumer Wastes

All the waste generated during the life cycle of dry legumes was collected in containers of different colors based on the municipal solid waste collection process, as follows:-Packaging wastes generated during production, storage at retailers, and consumer use (namely, PE super-sacks and PP bags, cardboard collars, labels, brass rivets, cartons, scotch tapes, PE shrink films, and broken semi-pallets) were collected in containers for plastic, paper and cardboard, or metal waste.-Dry legume wastes resulting from cleaning and packaging were pelletized, collected in the same PE super-sacks previously used for storing selected legumes, and delivered to local livestock farms, while the dust and herbaceous fractions recovered during the cleaning phase were collected and returned to the agricultural soil.-Cooked legume waste was discarded in containers for organic waste collection.

Packaging waste was disposed of according to the Italian scenarios for overall urban solid waste management in 2020 [64], as reported in Table 7. In 2019, 31% of the organic fraction was landfilled, 18% was incinerated, and 51% was recycled [65,66]. As suggested by EPD^®^ [67], it was assumed that 25.5% of the recycled fraction was composted and the remaining 25.5% underwent anaerobic digestion.

By referring to the recently updated percentage waste from the USDA Economic Research Service [68], the loss of cooked legumes during consumption was assumed to be 10% of the quantity served at the table. Such percentage waste was not only employed to assess the environmental impact from cradle to grave of legume production and consumption in the United States [38], but also cited in the food waste report by the Barilla Foundation [69].

The wastewaters withdrawn after chickpea and bean soaking, as well as the cooking of the three legumes under consideration, are typically drained into kitchen sinks. Their volumes were estimated by subtracting the water absorbed by the rehydrated and/or cooked legumes and evaporated during cooking (set as 5% of the initial water quantity) from the initial quantity of the soaking and cooking water used, as detailed in Table 6 and Table A3 in Appendix B.

Appendix A describes the inventory associated with the use phase.

To account for the end of life of the primary, secondary, and tertiary packaging material wastes associated with the functional unit chosen, the waste scenarios of plastic, paper and carboard, and non-ferrous metal wastes were allocated on a mass-based criterion, as reported in Appendix A. The disposal scenarios of primary, secondary, and tertiary packaging wastes are, respectively, shown in Appendix A, whereas Appendix A illustrates the disposal scenarios for combined packaging wastes. In particular, the end of life of the PP semi-pallet included 99.8% pallet reuse and 0.2% pallet disposal (Appendix A). In these stages, the final transport of packaging wastes to the Waste Collection Center was included.

#### 3.2.9. Life Cycle of Dry Legumes

The life cycle of dry legumes linked the assembly of dry legumes to their distribution logistics and consumer use, as well as the life cycles of the primary, secondary, and tertiary packaging materials, as shown in Appendix A. The latter is described in Appendix A and was linked to the life cycle of the PP semi-pallet, as detailed in Appendix A. Both these life cycles allowed their relative assembly stages to be related to their corresponding end-of-life disposal scenarios (Appendix A).

#### 3.2.10. Malting Process of Dried Legumes

The malting process of the three legumes under investigation was developed at the laboratory scale [17,18] and then transferred to the 100 kg/cycle pilot maltster (BBC Srl, Possagno, Italy) by carrying out the following three different steps:(a)Soaking at 25 °C for 3 or 5 h in the case of Onano lentils or Gradoli Purgatory beans and straight furrow chickpeas.(b)Germination at 25 °C for 72 h for any legume variety.(c)Drying at a maximum temperature of 60 °C for 12 h when processing 50 kg of dried legumes/cycle.

The malting tests carried out on the pilot plant scale treated lots of 50 kg of dried legumes per cycle, and yielded a specific consumption of approximately 8 L of process water (that is, 3 L/kg for the preliminary washing, 4 L/kg for the soaking step, and 1 L/kg to assure the appropriate moisture level during the germination step) and 0.8 kWh of electricity per kg of dried legume. Each kg of dried legume as-is gave rise to approximately 0.86 ± 0.2 kg of malted and decorticated Gradoli purgatory beans, 0.855 ± 0.15 kg of malted and decorticated straight furrow chickpeas, and 0.85 ± 0.2 kg of malted and decorticated Onano lentils [32]. On average, the malting conversion yield was approximately 0.853 ± 0.013 g/g, equivalent to a ratio between the legume as-is and malted legume of 1.17 ± 0.02 g/g. Such yields were found to be in line with those of the barley malting process, which has an average duration of 9 days and consists of a soaking step of about 48 h, a germination one of 96 h, and a kilning one of 24 h, followed by the separation of rootlets and calibration. Approximately, from 120 to 130 kg of calibrated barley is converted into 100 kg of barley malt depending on the quality and cleanliness of the seeds, this involving an average barley-to-malt ratio of 1.267 g/g. The overall water consumption is 7 L/kg of barley, while the overall energy consumption amounts to 0.88 kWh/kg of barley [70], including 0.75 kWh/kg of thermal energy (99% attributable to the kilning step) and 0.13 kWh/kg of electrical energy (29–30% attributable to the germination step, 40–45% to the kilning one, 12–15% to refrigeration during the soaking and germination steps, and 14–15% for the handling and cleaning/calibration of malted grains) [71].

Appendix A shows a block diagram of the malting process of selected grain pulses, which are then submitted to the same packaging process described in Figure 3 for untreated dry legumes. Appendix A shows the material balance of the malting, cleaning and dehulling, and packaging processes of malted and hulled legumes, including the management of the organic and packaging wastes formed. The logistics of input and output resources and product distribution, the energy sources used, the consumption phase, and the waste disposal scenario aligned with those described above for raw dried pulses.

### 3.3. Impact Assessment

The impact assessment was carried out using the Product Environmental Footprint (PEF) standard method [21], which was embedded in the software SimaPro 9.5.0.0 (PRé Consultants, Amersfoort, The Netherlands). In this way, 16 mid-point impact categories were estimated using the models reported below: *climate change* (CC) with the Bern model and 100-year time horizon Global Warning Potentials [72]; *ozone depletion* (OD) with the EDIP model and Ozone Depletion Potentials [73]; *ionizing radiation* (IR) with the Human Health effect model [74]; *photochemical ozone formation* (PhOF) with the LOTOS-EUROS model [75]; *particulate matter* (PM) with the UNEP model [76]; *acidification* (A) with the Accumulated Exceedance model [77]; *freshwater eutrophication* (FWE) and *marine eutrophication* (ME) with the EUTREND model [78]; *terrestrial eutrophication* (TE) with the Accumulated Exceedance model [77]; *freshwater eco-toxicity* (FWET), *non-cancer human toxicity* (NC-HT), and *cancer human toxicity* (C-HT) with the USEtox model [79]; *land use* (LU) with the LANCA^®^ v 2.2 baseline model [80]; *water use* (WU) with the Available Water Remaining (AWARE) model [81]; and *resource use-fossils* (RUF) and *resource use-mineral and metals* (RUMM) with the CML2002 model [82]. Owing to the international nature of supply chains, the European PEF standard method normalized each of the above mid-point impact categories with respect to their corresponding global impact [83]. Then, the normalized scores were weighted [84] and summed up to yield the so-called Overall Weighted Sustainability Score (OWSS). Owing to their limited resilience, the impact categories of human and eco-toxicity were excluded from the estimation of the OWSS. This calculation relied on the weighting coefficients proposed by Sala et al. [84] and adhered to the guidance outlined in the Product Environmental Footprint (PEF) category rules for dry pasta [60].

### 3.4. Sensitivity Analysis

Upon the integration of triangular and/or normal distribution uncertainty ranges associated with key agricultural management practices, pulse grain yields, and cleaning waste fractions into the LCA model, the application of the well-established Monte Carlo approach, as elucidated by Theodoris [85], became feasible. This analytical process was seamlessly incorporated into the LCA software SimaPro, employed for the current study, and involved the generation of random variables for each parameter characterized by the specified uncertainty range. Subsequently, the impact categories (ICs) and Overall Weighted Sustainability Scores (OWSS) underwent iterative recalculation and storage across 2000 repetitions of the procedure. The resulting diverse array of output values significantly contributed to the establishment of an uncertainty distribution.

The sensitivity of the OWSS was also assessed by selecting specific mitigation options.

## 4. Results and Discussion

### 4.1. Environmental Profile of Harvested Pulse Seeds at the Farm Gate

Considering the inputs of fertilizer (Fert), pesticide (Pest), seed density (Sd), and diesel fuel and lubricant oil (DFLO), Table 2 provides yield factors for both above- and below-ground biomasses, while Appendix A and Table 5 encompass on-field emissions (FE) from fertilized soil and transportation, respectively. These data allowed the mid-point environmental profile for each kilogram of the three pulse grains at the farm gate to be calculated using the PEF standard method, as shown in Table 8. Precisely, the Monte Carlo method employed for assessing the uncertainty range of each impact category demonstrated statistically significant differences at the 95% confidence level. Notably, the use of diesel fuel and lubricant oil in agricultural management practices exerted a considerable influence on various impact categories (specifically PhOF, PM, A, ME, TE, and RUF) across all three studied pulses. Additionally, for GPBs and OLs, CC also played a significant role, while WU emerged as a relevant factor for GPBs exclusively. On-field greenhouse gas emissions primarily impacted land use (LU), with contributions from CC or WU in the cases of SDCs or GPBs, respectively. Concerning climate change, the global warming potentials of SDC, GPB, or OL dried seeds at the farm gate were determined to be 0.59, 0.73, and 0.57 kg CO_2e_/kg, respectively. These values generally exceeded those reported by Bandekar et al. [38] for conventionally cultivated chickpeas, beans, or lentils in the USA, ranging from 0.39 to 0.61 or 0.45 kg CO_2e_/kg. Additionally, they were higher than the figures provided by Borghi et al. [37] for conventionally cultivated chickpeas (0.44 kg CO_2e_/kg) and beans (0.58 kg CO_2e_/kg) in Italy. This discrepancy can likely be attributed to the higher crop yields achieved in the referenced studies, specifically 1.8 [38] compared to 2.0–2.2 [37] Mg/ha for chickpeas, 1.9 [38] compared to 2.0–2.5 [37] Mg/ha for beans, and 1.3 Mg/ha for lentils [38].

The radar chart depicted in Figure 4 provides a comparative analysis of the mid-point impact categories for 1 kg of dry pulses at the farm gate in relation to dried Solco Dritto chickpeas. Notably, it highlights the elevated score in the water use category for Gradoli Purgatory beans (GPBs), attributed to their cultivation requiring irrigation (cf. Table 2). Additionally, the chart indicates lower impacts on specific categories (specifically RUMM, OD, IR, and FEW) for Onano lentils (OLs), primarily influenced by the reduced use of fossil-derived fertilizers in their cultivation (cf. Table 2).

For instance, the Sankey diagram [86] illustrated in Figure 5 highlights the significant contribution of input resources to the OWSS for 1 kg of *Solco Dritto* chickpeas at the farm gate. This emphasis is determined by the width of the arrows, which shows the relevant contributions of the use of seeding and diesel fuel to the flow of the OWSS.

As indicated in Table 8, the freshwater eco-toxicity (ETFW) and human-toxicity (NC-HT and C-HT) impact categories, particularly notable in the case of OLs, displayed, as anticipated, low robustness, and were therefore excluded from the assessment of the Overall Weighted Sustainability Score (OWSS). In light of this, the remaining 13 impact categories (ICs) underwent normalization and were subsequently multiplied by a set of weighting factors, representing the perceived relative importance of the considered life cycle impact categories, as detailed in Table 9. It is crucial to highlight that weighting is an essential step in PEF studies, aiding the interpretation and communication of analysis results and facilitating the comparison of weighted outcomes across different impact categories to gauge their relative significance. Furthermore, these weighted results may be aggregated across life cycle impact categories to derive a singular overall score.

The land use impact category (LU) played a predominant role, contributing 52% and 59% to the Overall Weighted Sustainability Score (OWSS) for Solco Dritto chickpeas (SDCs) and Onano lentils (OLs), respectively. However, for Gradoli Purgatory beans (GPBs), LU emerged as the secondary contributor due to heavy reliance on irrigation in their cultivation, with the water use impact category making the primary contribution at approximately 61% of the OWSS. Across all three pulse varieties studied, the climate change impact category (CC) resulted in being the second- or third-most significant factor, with its contribution ranging from 16.7% to 14.1% and 6.1% for SDCs, Ols, and GPBs, respectively.

Figure 6 enables a clear evaluation of the distinct contributions made by the impact categories considered in the PEF standard method.

### 4.2. Cradle-to-Grave Environmental Profile of Dry Pulses

Table 10 presents the business-to-consumer (B2C) midpoint impact category scores for one functional unit of the considered dry pulses. It includes primary (PHS) and secondary (SHS) hotspots, along with their respective percentage contributions in brackets.

Throughout the field phase of all dry pulses, the impact categories that experienced the most pronounced effects were land use (97–98%), freshwater eco-toxicity (90–95%), resource use-fossils (76–85%), terrestrial eutrophication (54–65%), particulate matter (47–60%), photo-chemical ozone formation (42–55%), acidification (38–48%), and non-cancer human toxicity (37–44%). The water use impact category was also influenced in specific instances, such as with SDCs (37%) and GPBs (99%). Climate change and marine eutrophication were impacted by the field phase at rates of 33% and 64% for GPBs and 36% and 65% for Ols, respectively. The consumer use phase of all dry pulses affected ozone depletion (37–44%) and ionizing radiation (30–33%). Additionally, SDCs experienced a 29% impact on climate change, possibly due to longer cooking times, while OLs faced a 33% impact on water use. Particulate matter had a predominant effect on freshwater eutrophication (28–30%) and resource use-minerals and metals (44–48%). Transportation played a significant role in cancer human toxicity (47–50%) for all examined pulses.

Figure 7 illustrates a comparative analysis of the mid-point impact categories for 1 kg of dry pulses in a cradle-to-grave perspective, specifically focusing on dried Solco Dritto chickpeas. Notably, the radar chart highlights a significant relative score in the water use category for GPBs, attributable to their irrigated field practices (refer to Table 2). Additionally, the chart reveals higher impacts on eutrophication freshwater (ETFW) and land use (LU) for OLs, likely stemming from their lower harvested and cleaned grain yields, as outlined in Table 2 and Table 3.

Table 10 shows that the cradle-to-grave global warming potentials of SDC, GPB, and OL dried seeds amounted to 2.5 ± 0.1, 2.6 ± 0.2, and 2.6 ± 4.5 kg CO_2e_/kg, respectively. On one hand, the identified values closely aligned with the lower threshold value documented by Bandekar (2022) [38] for a 60 g portion of pulses manufactured and utilized in the United States, which ranged from 0.12 to 1.34 kg CO_2e_/portion. It is important to highlight that the estimates provided by Borghi et al. [37], specifically 0.97 or 1.17 kg CO_2e_ per kg of chickpeas or Borlotti beans when packaged in glass bottles or steel tin cans, did not encompass the consumer use phase.

As an illustration, the Sankey diagram in Figure 8 portrays the diverse contributions from each life cycle stage to the overall Weighted Sustainability Score (OWSS) for 1 kg of Solco Dritto chickpeas, spanning from the field phase to post-consumption waste disposal. The width of the arrows visually signifies the substantial impacts of factors like the field and consumer use phases on the OWSS flow.

Table 11 illustrates the end-point characterization of the business-to-consumer (B2C) environmental profile of dry pulses in accordance with the PEF method.

The land use impact category (LU) played a crucial role, contributing 28.5% and 39.2% to the Overall Weighted Sustainability Scores (OWSS) for Solco Dritto chickpeas (SDCs) and Onano lentils (OLs), respectively. However, for GPBs, LU assumed a secondary role, with its percentage contribution decreasing to 17.6% of the OWSS due to irrigated cultivation, making the water use impact category the primary contributor at approximately 49% of the OWSS.

Across all three pulse varieties studied, the climate change impact category (CC) emerged as the second- or third-most significant contributor, with contributions ranging from 28.57% for SDCs to 22.9% for OLs and 13.1% for GPBs, respectively. Particulate matter (PM) followed, with contributions of 8.7%, 4.7%, and 8.8%, respectively.

Figure 9 visually highlights the diverse contributions made by the impact categories considered in the PEF standard method.

Ultimately, the B2C Overall Weighted Sustainability Score (OWSS) was nearly 254, 581, and 329 µPt for 1 kg of SDCs, GPBs, and OLs produced and consumed in the Latium region of Italy, respectively. These scores were primarily influenced by the agricultural phase (56–79%), followed by transportation (6.3–12.5%), and then packaging materials (4.9–10.1%). Notably, consumer use emerged as the second-most significant contributor to the OWSS, making up 12.8% for SDCs. This was likely attributed to their prolonged cooking times in comparison to beans and lentils (cf. Table 6).

It can be also noted that Table 11 highlights a substantial range of uncertainty in the impact category scores, particularly noticeable in the case of OLs. This variation was directly attributed to the significantly wide range of cleaning waste percentages observed for OLs, spanning from as low as 25% to as much as 40% of the harvested lentil seeds delivered to the processing facility. In contrast, SDCs and GPBs presented a more constrained range of cleaning waste, fluctuating from 10% to 15% (cf. Table 3). The maximum cleaning waste figures for OLs were observed in recent harvesting campaigns, where unfavorable conditions of high temperatures and drought adversely impacted lentil crops, resulting in elevated percentages of empty seeds. Optical sorters rejected these empty seeds, leading to a notable increase in waste percentage due to discarding.

In the LCA models used thus far to assess the environmental impact of the three examined legumes, the cultivation processes of the seeds were based on the data available in the Agri-footprint v. 6.3 database, integrated into the LCA software SimaPro. However, it should be noted that, at the reference farm, the seeds used were produced on-site and constituted a portion of the seeds previously fumigated and selected for subsequent packaging once their germination had been verified. The introduction of this option in the aforementioned LCA models implied an internal loop, as highlighted in the Sankey diagram related, for instance, to Solco Dritto chickpeas and illustrated in Figure 10.

Table 12 provides the effective and normalized and weighted values for various impact categories, along with Business-to-Consumer (B2C) Overall Weighted Sustainability Scores (OWSS). Notably, the adoption of in situ cultivated seeding material had a relatively modest impact on the average OWSS values. However, it is noteworthy that the substantial coefficient of variation observed earlier in the case of OLs (refer to Table 10), likely stemming from the chosen lentil seed cultivation process (specifically, lentils, start material, at seed production {AU} mass), was significantly reduced to 10%.

Even in this scenario, the land use impact category (LU) played a primary role, contributing 31% and 42% to the Overall Weighted Sustainability Score (OWSS) for SDCs and OL, respectively. However, for GPBs, LU assumed a secondary role with a contribution of 17% to the OWSS, primarily due to irrigated cultivation, where the water use impact category became the primary contributor at approximately 51% of the OWSS.

The climate change impact category (CC) emerged as the second- or third-most significant contributor, with contributions ranging from 28% for SDCs to 21% for OLs and 12% for GPBs, respectively. Particulate matter (PM) followed, with contributions of 9% for SCDs and OLs and 48% for GPBs. Finally, when using in situ cultivated seed material, the B2C Overall Weighted Sustainability Scores for 1 kg of SDCs, GPBs, and OLs produced and consumed in Italy’s Latium region were not significantly different from those estimated using seeds from external sources (Table 11), equating to 259 ± 25, 624 ± 99, and 349 ± 35 µPt, respectively.

The assessment of the malting process’s impact on the three dried pulses will be conducted with reference to in situ cultivated seeding material.

### 4.3. Cradle-to-Grave Environmental Profile of Dry Malted and Decorticated Pulses

Table 13 presents the effective values, as well as the normalized and weighted values, for various impact categories, accompanied by the B2C Overall Weighted Sustainability Scores (OWSS) related to the production and consumption of malted and decorticated pulses in 500 g PP bags.

Despite the malting process leading to higher rates of process water and electricity consumption and generating additional discarded materials like cuticles and rootles, the overall scores for the malted and decorticated pulses increased by 15–17% compared to those of dry pulses alone. These scores ranged from 298 ± 30 to 410 ± 40 and 731 ± 113 µPt for 1 kg of malted and decorticated Solco Dritto chickpeas (MDSDC), Onano lentils (MDOL), and Gradoli Purgatory beans (MDGPB), respectively. In terms of global warming potentials, the malting process resulted in an increase in greenhouse gas (GHG) emissions to 2.8 and 3.0 kg CO_2e_ per kg of MDSDCs, MDGPBs, and MDOLs, respectively. This increase ranged from 13.3% to 16.7% compared to the untreated dry GPBs and SDCs and OLs seeds. Even for these malted products, the land use impact category (LU) remained a predominant factor, contributing 31% and 42% to the OWSSs for MDSDCs and MDOLs, respectively. As expected, in the case of MDGPBs, the water use impact category retained its status as the primary contributor, accounting for circa 52% of the OWSS, with land use (LU) following with an 18% contribution to the OWSS. The climate change impact category (CC) was the second contributor, with contributions ranging from 28% for SDCs to 22% for OLs, but the third contributor with 12% of the OWSS for GPBs. The contribution of particulate matter (PM) ranged from 9% for SCDs and OLs to 4% for GPBs.

Figure 11 visually highlights the diverse contributions made by the impact categories considered in the PEF standard method, as well as the cradle-to-grave Overall Weighted Sustainability Score (OWSS) for 1 kg of malted and decorticated pulses.

### 4.4. Options to Improve the Sustainability of Dry Pulses as Such and Malted and Decorticated

Mitigation strategies for dry pulses, whether in their natural state or malted and decorticated form, should prioritize addressing the primary impact stemming from the agricultural phase, particularly concerning land and water utilization. Moreover, there is a need for a secondary focus on mitigating the effects of climate change and its closely related impact categories, which include acidification, eutrophication, photochemical ozone formation, and the use of fossil resources [87,88].

#### 4.4.1. Land Use Mitigation

The substantial impact of LU on the OWSS in the case of chickpeas and lentils is undoubtedly attributed to the previously mentioned low crop yields observed in the typical areas of their cultivation (see Table 2). Indeed, growing identical varieties of pulses in a neighboring area, situated approximately 40–60 km away and thus beyond the designated cultivation zone, in the municipalities of Canino (Lat. 42.466394; Long. 11.750672) and Montalto di Castro (Lat. 42.351566; Long. 11.607010) at average altitudes of 229 and 42 m above sea level, respectively, yielded crops ranging from 2.3 to 3.0 Mg/ha for SDCs and 1.5 to 2.0 Mg/ha for OLs [51]. The identical fertilization and management practices outlined in Table 2 were implemented in both locations.

In the case of SDCs, this increase in crop yield from 1.71 to 3 Mg/ha resulted in a decrease in the effective score of the land use impact category from 910 ± 150 to 487 ± 26 Pt/kg and climate change from 2.8 ± 0.2 to 2.42 ± 0.07 kg CO_2e_/kg. When considering the normalized and weighted scores, the impact of land use decreased from 93 ± 16 to 50 ± 3 μPt/kg, and the impact of climate change from 82 ± 5 to 71 ± 2 μPt/kg. Consequently, the Overall Weighted Sustainability Score (OWSS) experienced a reduction of approximately 25%, declining from 298 ± 30 to 221 ± 7 μPt/kg. This outcome underscores the substantial impact of crop yield on the overall sustainability of chickpea cultivation.

#### 4.4.2. Water Use Mitigation

Considering the heightened sensitivity of GPBs to the water use impact category, it is well-established that drought stress imposes a substantial constraint on the growth and productivity of common beans [89,90]. Extensive research has been conducted in Latin America, Africa [91], and Asia [92], with a focus on enhancing drought tolerance and improving the production of common beans through the selection of various physiological and genetic traits. Consequently, drought tolerance emerges as a pivotal trait in the selection of common bean varieties for production in regions prone to drought stress, attributable to factors such as diminishing water supplies and climate change [93]. For example, red beans have shown commendable adaptation to restricted water supplies and prolonged dry summers in the central highlands of Afghanistan [92]. In such a scenario, choosing a drought-tolerant variant of Gradoli Purgatory beans tailored for rainfed conditions would lead to a noteworthy decrease in their Overall Weighted Sustainability Score, as illustrated in Figure 12. Notably, the normalized and weighted scores for the water use, land use, and climate change impact categories decreased from 378, 128, and 90 to 3, 126, and 54 μPt/kg, respectively. Consequently, the OWSS stabilized from 731 ± 113 to 264 ± 45 μPt/kg, affirming the substantial impact of water use on the overall sustainability of bean cultivation.

#### 4.4.3. Climate Change Mitigation

To mitigate the impact of the climate change category, several actions can be pursued:Minimizing electricity consumption in the malting process involves two key considerations. First, it is crucial to validate the specific consumption rate of 0.8 kWh per kg of dry pulses during malting, as determined in a pilot-scale malter, for its accurate application in industrial-scale equipment. Second, exploring the potential adoption of solar tunnel dryers for the kilning step represents a promising avenue for further reducing environmental impact.Transitioning to photovoltaic electricity: the company relied solely on electricity for dry legume production, consuming 32,900 kWh from the Italian medium-voltage grid in 2022 [51]. After installing photovoltaic panels on warehouse roofs, approximately 19,000 kWh per year was generated. Half of this was used on-site, contributing to about 29% of the total annual electricity consumption [51]. Expanding this photovoltaic paneling could potentially cover the entire factory’s electricity needs with solar power.Upgrading transport vehicles: replace the current light commercial vehicles, used for transporting the majority of resources in legume cultivation, packaging, and distribution, with new 1200 kg diesel vans complying with the CO_2_ emission performance target (95 g CO_2e_/km) set by the European Community in 2019 [94]. This change would reduce the emission factor from 2.01 to just 0.079 kg CO_2e_ Mg^−1^ km^−1^.Optimizing pulse cooking with energy-efficient appliances: transition from gas-fired and electric kitchen appliances to energy-efficient home appliances, such as induction-heated cookstoves, to optimize pulse cooking. According to Table A3, the energy consumption during pulse cooking was minimal with induction cookstoves, ranging from 1.15 to 0.99 and 0.70 kWh per kg for dried SDCs, GPBs, and OLs, respectively. For malted and decorticated pulses, the use of induction hobs could further reduce energy needs from 0.85–1.12 (Table 6) to 0.62–0.78 kWh per kg. This transition aligns with sustainability goals by minimizing energy consumption.

While the economic viability of transitioning to photovoltaic electricity (item ii) and upgrading transportation vehicles (item iii) may seem feasible even for small-scale producers, such as the target factory considered in this study, their successful implementation is contingent upon the specific economic circumstances of the entities involved. Furthermore, item iv focuses on governmental incentives to replace traditional cookstoves with smart alternatives, coupled with detailed guidance on achieving optimal food boiling with minimal energy consumption. Implementing such options would impact the Overall Weighted Sustainability Score (OWSS), taking into account that the climate change impact category (CC) contributed 28% for SDCs, 22% for OLs, and 12% for GPBs, as indicated in Table 13.

## 5. Conclusions and Future Perspectives

The environmental impact of innovative malted and decorticated pulses, specifically *Solco Dritto* chickpeas (SDC), *Gradoli Purgatory* beans (GPB), and *Onano* lentils (OL), native to the Latium region of Italy, was investigated in a business-to-business context. These pulses, characterized by a low phytate and virtually zero oligosaccharides content, and shorter cooking times compared to their conventional dried counterparts, were analyzed using a Life Cycle Assessment (LCA) approach. Utilizing a widely recognized LCA software SimaPro and the Product Environmental Footprint standard method, the study estimated n overall carbon footprints of 2.8 and 3.0 kg CO_2e_ per kg of malted and decorticated SDCs and GPBs and OLs, respectively. This represented a 13–17% increase compared to untreated dry seeds. A similar increase was observed for the Overall Weighted Sustainability scores (OWSS), ranging from 298 ± 30 to 410 ± 40 and 731 ± 113 µPt/kg for MDSDCs, MDOLs, and MDGPBs, respectively. The predominant factor influencing the OWSS was found to be the land use impact category (LU), contributing 31% and 42% to the OWSS for MDSDCs and MDOLs, respectively. In the case of MDGPBs, water use impact accounted for about 52% of the OWSS, while LU represented 18%. The agricultural phase played a pivotal role in the environmental impact of dry pulses, whether in their original state or malted and decorticated. The climate change impact category (CC) was the second-largest contributor, ranging from 28% (MDSDCs) to 22% (MDOLs), and ranking as the third contributor with 12% of the OWSS for MDGPBs. Mitigation actions should primarily focus on reducing the impact of the agricultural phase, especially improving land and water utilization. Selecting highly productive and drought-resilient pulse varieties could significantly reduce the OWSS by up to 64% in the case of drought-tolerant bean varieties. To mitigate the impact of climate change, suggested actions include optimizing electricity consumption during malting, transitioning to photovoltaic electricity, and reducing greenhouse gas emissions from transport vehicles. Additionally, optimizing pulse cooking with energy-efficient appliances could further enhance sustainability.

These initiatives, aligned with sustainability objectives, may promote the utilization of malted and decorticated pulses in gluten-free, low-fat, α-oligosaccharides, and phytate-specific food products tailored for individuals with celiac disease, diabetes, and hyperlipidemia. However, nutritionally and environmentally conscious consumers should recognize that the energy expended during the malting process, along with the generation of additional processing waste (radicles and cuticles), has led to a 13–17% increase in carbon footprint and Overall Weighted Sustainability Score compared to those of whole dry legumes.

In conclusion, this holistic approach tackles environmental concerns and advocates for sustainable practices, fostering innovation in pulse utilization to enhance dietary options. However, further research is required to assess consumer acceptability and willingness to pay for improved dietary choices.

## Figures and Tables

**Figure 1 foods-13-00655-f001:**
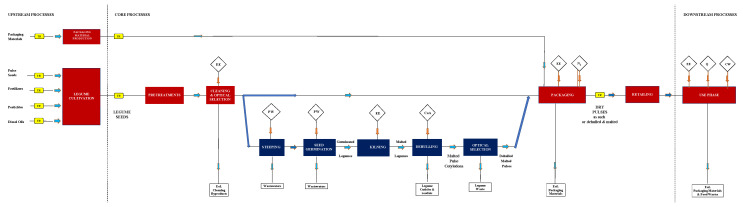
System boundary of the production and consumption system of dried pulses as such or malted and decorticated, where the main processes are subdivided into upstream, core, and downstream ones: CoA, compressed air; CW, cooking water; EE, electric energy; EoL, end of life; N_2_, gaseous nitrogen; PW, process water; Q, thermal energy; and TR, transport.

**Figure 2 foods-13-00655-f002:**
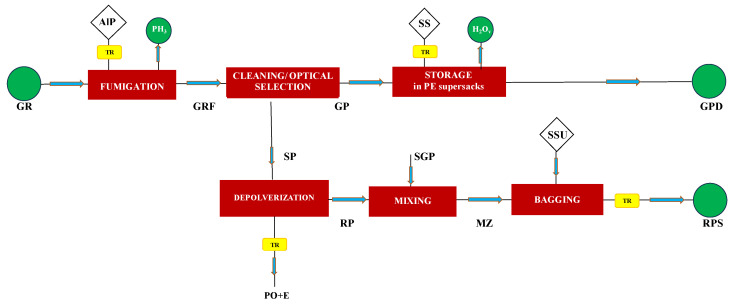
Block diagram of the fumigation process, cleaning, and optical selection of harvested grains, and storage of cleaned grains. For all symbols refer to the Nomenclature section.

**Figure 3 foods-13-00655-f003:**
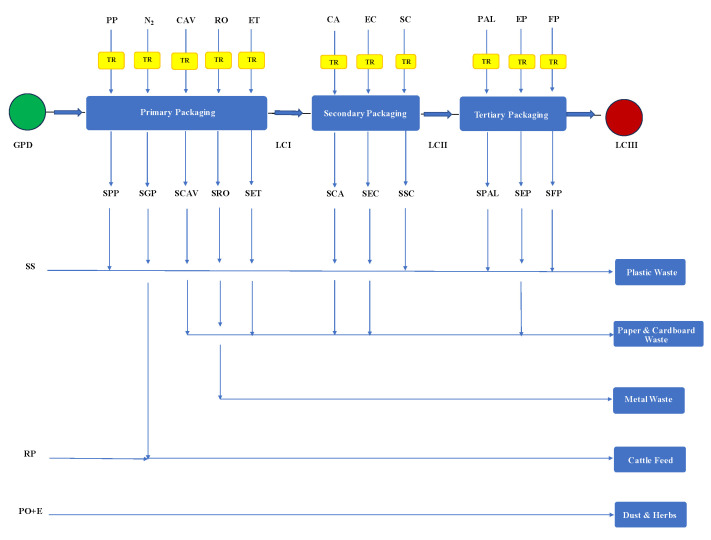
Flowchart of the packaging and organic and packaging waste management processes for dry legumes. For all symbols, refer to the Nomenclature section.

**Figure 4 foods-13-00655-f004:**
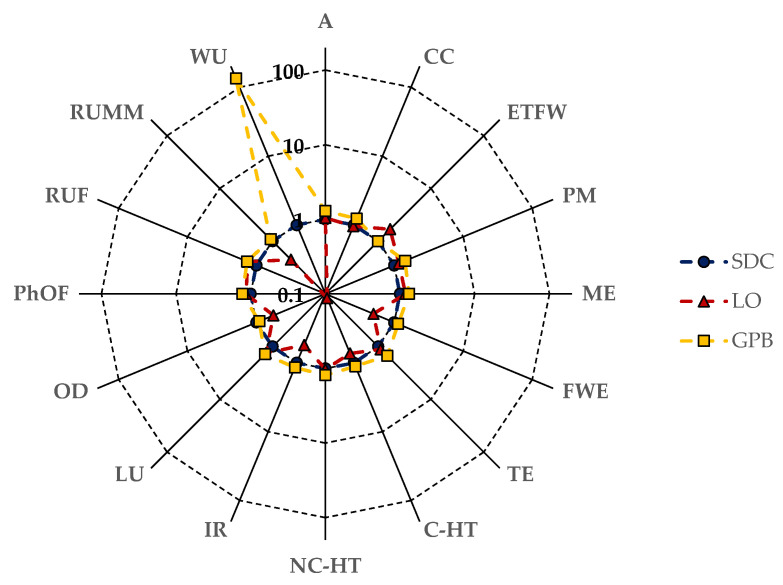
Radar chart comparing the mid-point impact categories for 1 kg of dry pulses at the farm gate with those of dried *Solco Dritto* chickpeas following the PEF standard method. Refer to the Nomenclature section for the symbols used to indicate each impact category.

**Figure 5 foods-13-00655-f005:**
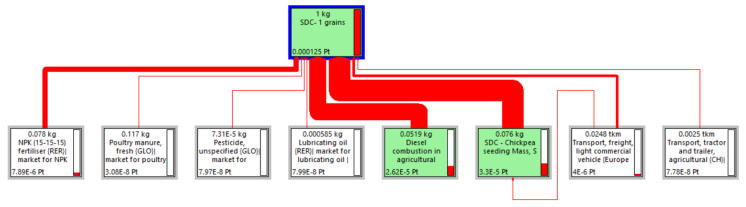
Sankey diagram depicting the contribution of the input resources to the Overall Weighted Sustainability Score for 1 kg of SDCs at the farm gate elaborated by the LCA software SimaPro using the PEF standard method and a cut-off percentage of 0.01%.

**Figure 6 foods-13-00655-f006:**
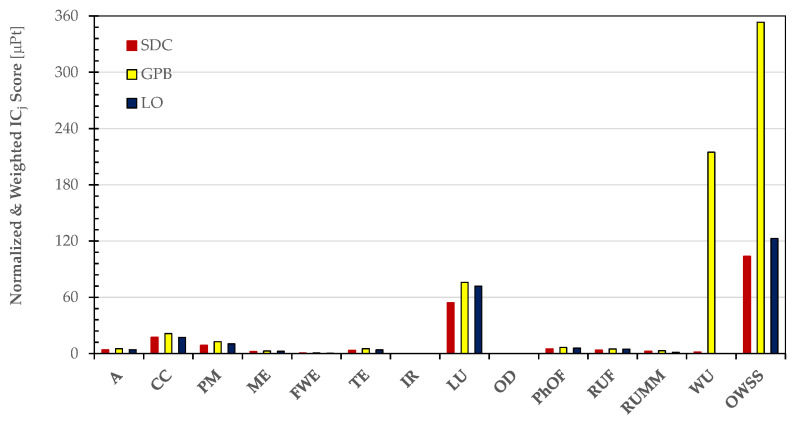
Normalized and weighted scores of the impact categories considered in the PEF standard method to estimate the Overall Weighted Sustainability Score (OWSS) for 1 kg of the dried pulses (SDC, GPB, and OL) at the farm gate.

**Figure 7 foods-13-00655-f007:**
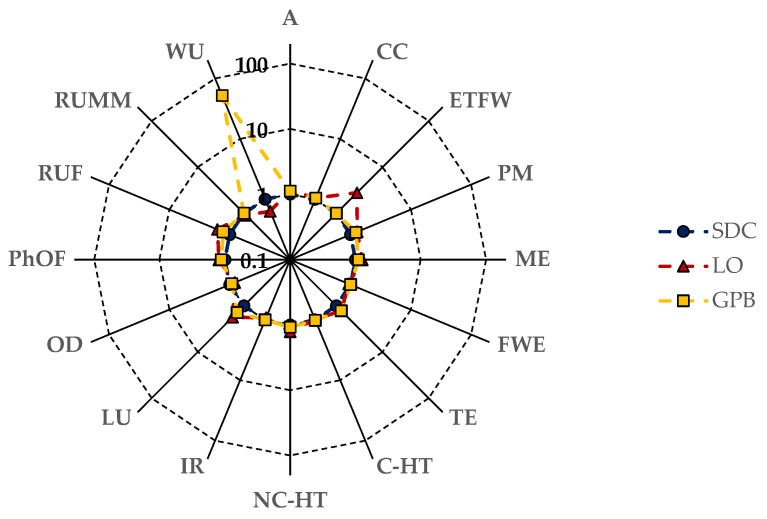
Radar chart comparison of cradle-to-grave scores for mid-point impact categories (IC) in 1 kg of dry pulses and dried Solco Dritto chickpeas using the PEF standard method. Please refer to the Nomenclature section for symbols representing each IC.

**Figure 8 foods-13-00655-f008:**
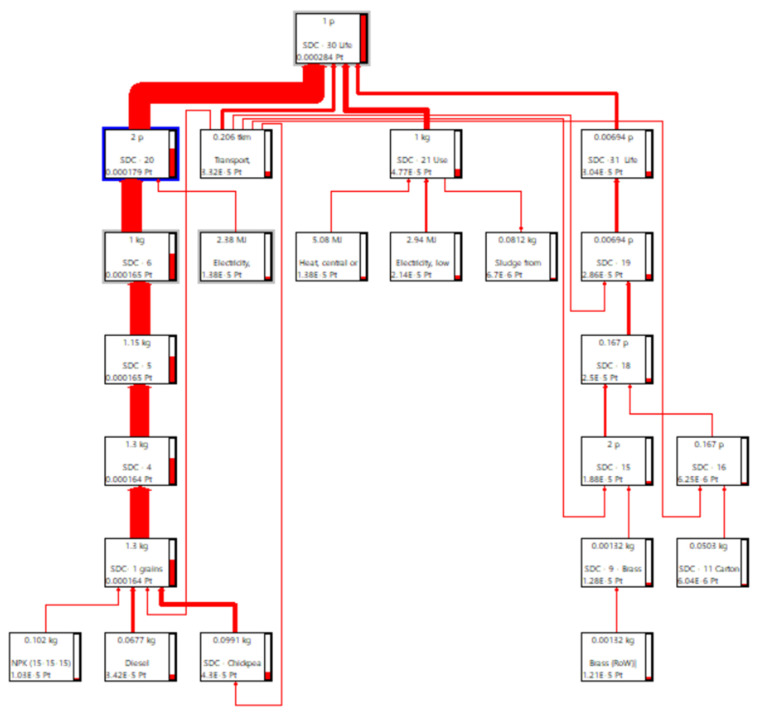
Sankey diagram depicting the contribution of the input resources to the cradle-to-grave Overall Weighted Sustainability Score for 1 kg of SDCs elaborated by the LCA software SimaPro using the PEF standard method and a cut-off percentage of 2%.

**Figure 9 foods-13-00655-f009:**
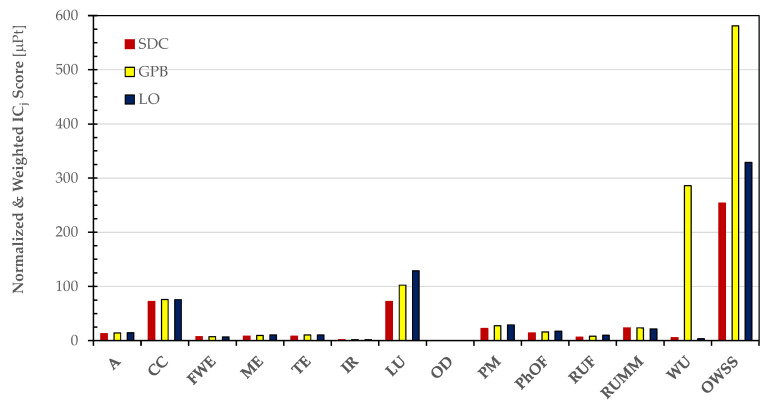
Normalized and weighted scores of impact categories in the PEF standard method used to estimate the cradle-to-grave Overall Weighted Sustainability Score (OWSS) for 1 kg of the dried pulses (SDC, GPB, and OL).

**Figure 10 foods-13-00655-f010:**
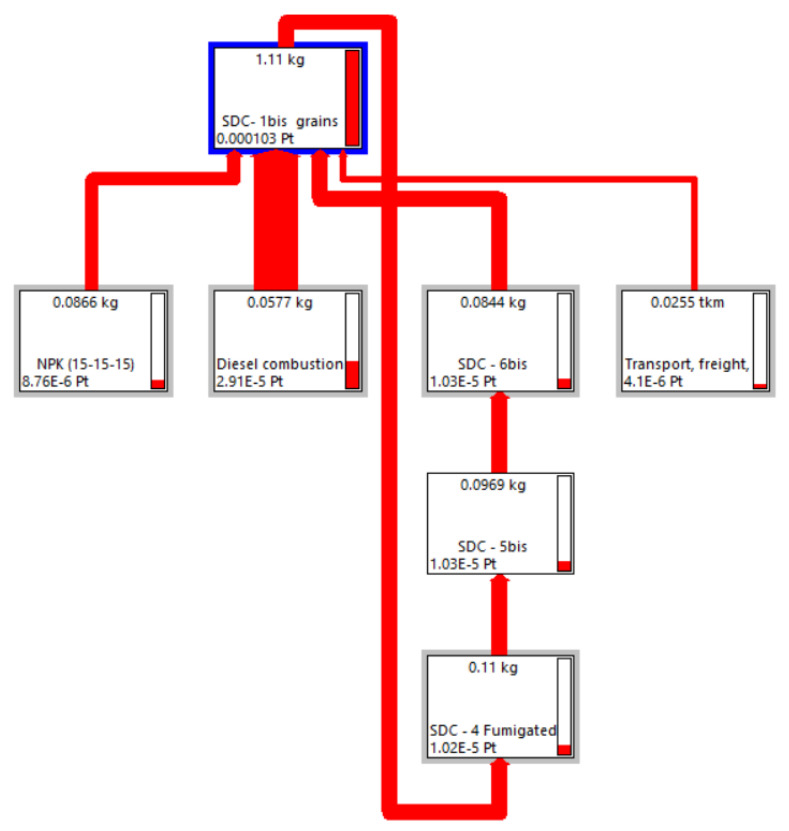
Sankey diagram illustrating the allocation of input resources and their impact on the Overall Weighted Sustainability Score for 1 kg of SDCs at the farm gate, particularly focusing on the use of in situ cultivated seeding material. This analysis was conducted through the LCA software SimaPro, utilizing the PEF standard method and a percentage cut-off of 0.1%.

**Figure 11 foods-13-00655-f011:**
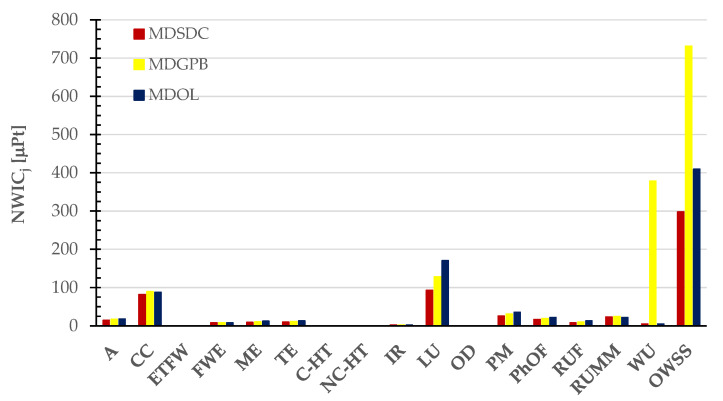
Normalized and weighted scores of impact categories in the PEF standard method used to estimate the cradle-to-grave Overall Weighted Sustainability Score (OWSS) for 1 kg of the dried malted and decorticated pulses (MDSDC, MDGPB, and MDOL).

**Figure 12 foods-13-00655-f012:**
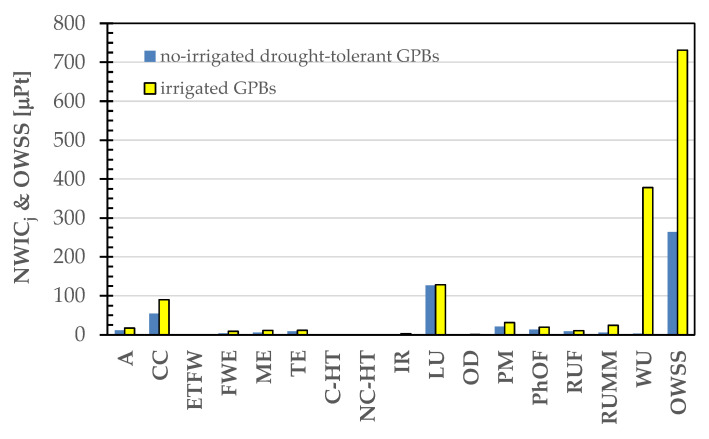
Comparative analysis of normalized and weighted impact category scores (NWIC_j_) and cradle-to-grave Overall Weighted Sustainability Score (OWSS) for 1 kg of dried malted and decorticated beans under irrigated and non-irrigated cultivation conditions in the PEF standard method.

**Table 1 foods-13-00655-t001:** Dry basis composition of Solco Dritto chickpeas, Gradoli Purgatory beans, and Onano lentils, as such (SDC, GPB, and OL) and malted and dehulled (MDSDC, MDGPB, and MDOL).

Component	SDC	MDSDC	GPB	MGPB	OL	MDOL	Unit
Raw Protein	22.3 ± 1.7	23.6 ± 1.9	22.7 ± 1.7	23.4 ± 2.1	26.1 ± 2.0	28.7 ± 2.2	g/100 g dm
Total Starch	46.8 ± 0.6	45.2 ± 2.0	33.81 ± 1.66	34.96 ± 0.19	50.9 ± 0.4	52.1 ± 2.8	g/100 g dm
Resistant Starch	1.77 ± 0.22	1.19 ± 0.43	23.59 ± 0.34	22.0 ± 1.8	2.30 ± 0.17	1.88 ± 0.47	g/100 g dm
Phytic Acid	1.15 ± 0.12	0.79 ± 0.09	1.15 ± 0.12	0.78 ± 0.13	1.09 ± 0.09	0.80 ± 0.02	g/100 g dm
Raffinose	3.80 ± 0.15	1.65 ± 0.11	5.31 ± 0.28	1.95 ± 0.20	3.78 ± 0.04	0.79 ± 0.07	g/100 g dm

**Table 2 foods-13-00655-t002:** Cultural conditions used for growing the three varieties of legumes examined here (SDC, GPB, and OL).

Legume	SDC	GPB	OL	Unit
Overall cultivation area	10	7–10	30	ha
Fallow area	0	0	0	ha
Town	Acquapendente (Italy)			-
Latitude	42.73661556684243			DD
Longitude	11.891832053277597			DD
Altitude	438			asl m
Minimum grain yield	1.11	0.78	0.93	Mg/ha
Maximum grain yield	2.35	1.76	1.67	Mg/ha
Average grain yield	1.71	1.26	1.26	Mg/ha
Above ground biomass use	100% left in the field after shredding			%
Below-ground biomass use	100% left in the field after shredding			%
Seed density	130	100	100	kg/ha
20-20-20 fertilizer	100	100	20	kg/ha
4-4-4 poultry manure	200	200	200	kg/ha
Herbicide (Feinzin)	0.125	0.125	0.125	kg/ha
Irrigation water withdrawn from the *Val di Paglia* Consortium	-	700–800	-	m^3^/ha
Overall diesel fuel consumption	90–120			L/ha
Agricultural machinery lubricant oil consumption	1			L/ha
Storage	in loco			
Transportation mode	Tractor with a 10 Mg trailer			
Field-to-harvester distance	0–4			km
Pulse moisture content resulting from solar drying	0.12			g/g

asl, above sea level; DD, decimal degree.

**Table 3 foods-13-00655-t003:** Minimum, maximum, and average percentages of cleaning waste for the three legumes examined here (SDC, GPB, and OL) with their average composition and average cleaned grain yield per hectare.

	Legume	SDC	GPB	OL	Unit
Parameter					
Minimum–maximum cleaning waste range		10–15	10–15	25–40	%
Average cleaning waste		12.5	12.5	32.5	%
- Dust fraction		2	3	5	%
- Grass and insect fraction		6	5	15	%
- Broken legumes, etc., fraction		4.5	4.5	12.5	%
Average cleaned grain yield		1.5	1.1	0.85	Mg/ha

**Table 4 foods-13-00655-t004:** Dry pulse packaging: mass of any component of the primary, secondary, and tertiary packages used.

Packaging Type	Technical Specifications	Unit
Primary Packaging	PP bags	
Mass of dried pulses	500	g
Mass of a PP bag	5.0 ± 0.3	g
Width × Depth × Height	80 × 50 × 200	mm × mm × mm
Thickness	100	mm
Mass of cardboard collar	5.8	g
Mass of brass rivets	no. 2 × 0.33	g
Mass of adhesive paper label	0.323	g
Massa of gaseous N_2_	5.0	g
Primary packaging overall mass	516.5	g
Secondary Packaging	Cardboard carton	
No. of primary packages	12	-
Length × Width × Height	380 × 280 × 120	mm × mm × mm
Carton mass	302 ± 3	g
Mass of adhesive label for cartons	2.0	g
Mass of scotch tape	4.0	g
Mass of dry pulses per carton	6.00	kg
Secondary packaging overall mass	6.505	kg
Tertiary Packaging	PP Semi-pallet	
Semi-pallet mass	5	kg
Length × Width × Height	600 × 800 × 144	mm × mm × mm
No. of cartons per layer	4	-
No. of layers per pallet	6	-
Overall height of pallet	0.864	m
Paper label per pallet	no. 2 × (3.108 ± 0.05)	g
Stretch-and-shrink PE film	287	g
Mass of dried pulses per pallet	144	kg
Tertiary packaging overall mass	161.42	kg

**Table 5 foods-13-00655-t005:** Logistics of input/output materials with indication of the means of transport used with the corresponding load capacity and distance travelled from different production sites to destination ones. All symbols are listed in the Nomenclature section.

Input/Output Materials	From	To	Means of Transport	Load Capacity [Mg]	Distance [km]
Seeds	PS	Field	LCV	1.3	25
NPK Fertilizer	PS	Field	LCV	1.3	100
Poultry Manure	PS	Field	LCV	1.3	100
Herbicide	PS	Field	LCV	1.3	25
Diesel Fuel	PS	Field	LCV	1.3	25
Lubricant Oil	PS	Field	LCV	1.3	25
Fresh Pulse Grain	Field	FG	Tractor and Trailer	10	4
Cleaning Waste	FG	CaF	Tractor and Trailer	10	50
AlP	PS	FG	LCV	1.3	280
PE Super-sacks	PS	FG	LCV	1.3	355
Gaseous N_2_	PS	FG	Euro5 HRT	10	35
Cardboard Collars	PS	FG	LCV	1.3	30
Brass Rivets	PS	FG	LCV	1.3	200
Cartons	PS	FG	Euro5 HRT	13.9	105
Paper Labels	PS	FG	LCV	1.3	30
PP bags, PE films, Scotch Tape	PS	FG	Euro5 HRT	13.9	200
PP Semi-pallet	PS	FG	Euro5 HRT	13.9	200
Palletized Dried Pulses	FG	PoS	LCV	1.3	150
Dust and Herbs (PO + E)	FG	Field	Tractor and Trailer	10	4
Packaging or Organic Waste	FG, R or UH	WCC	MWCS	13.9	50

CaF, Cattle farm; FG, Factory gate; HRT, Heavy rigid truck; LCV, Light Commercial Vehicle; MWCS, Municipal Waste Collection Service; PoS, point of sale; PS, production site; R, retailer; UH, User’s house; and WCC, Waste Collection Center.

**Table 6 foods-13-00655-t006:** Operating mode for the soaking and cooking phases of the three legumes under study, both as such (AS) and malted and dehulled (MD): e_C_, specific electricity consumption; ECT, effective cooking time; LCT, cooking time on the label; ST, soaking time; and WPR, water-to-dry pulse ratio.

	Phase	Soaking				Cooking					
Legume		AS			MD		AS			MD	
			WPR *	ST		WPR ^#^	LCT	ECT	e_C_	ECT	e_C_
			[L/kg]	[h]		[L/kg]	[min]	[min]	[kWh/kg]	[min]	[kWh/kg]
SDC		Yes	4	16–24	No	4	90 ^Ɉ^	90	1.68	45 ^ɸ^	1.12
GPB		Yes	4	16–24	No	4	80 ^§^	60	1.42	30 ^ɸ^	0.98
OL		No	-	-	No	4	16 ^¥^	30 ^ɸ^	0.98	15 ^ɸ^	0.85

* [53]; ^#^ [54]; ^Ɉ^ https://espressotuscia.it/legumi-e-cereali/217-cece-del-solco-dritto-di-valentano-confda-500-g-azienda-agricola-cerqueto.html; ^§^ https://espressotuscia.it/legumi-e-cereali/214-fagiolo-del-purgatorio-di-gradoli-confda-500-g-azienda-agricola-cerqueto.html; ^¥^
https://espressotuscia.it/legumi-e-cereali/211-lenticchia-di-onano-confda-500-g-azienda-agricola-cerqueto.html (accessed on 12 February 2024); ^ɸ^ [31].

**Table 7 foods-13-00655-t007:** Overall Italian waste management scenarios for packaging and organic wastes in 2019 and 2020, as formed during the dry pulse processing, distribution, and consumer phases.

Waste Management Scenario	Landfill [%]	Recycling [%]	Incineration [%]	References
Non-ferrous metal waste	25.4	68.1	6.5	[64]
Organic waste	31	51	18	[65,66]
Paper and cardboard waste	5.2	87.3	7.5	[64]
Plastic waste	7.4	45.6	47.0	[64]

**Table 8 foods-13-00655-t008:** Environmental profile for 1 kg of the dried pulses (SDC, GPB, and OL) at the farm gate according to the PEF standard method: type of the primary hotspot (PHS) type and corresponding percentage contribution, and mean value and standard deviation for each mid-point impact category (ICj) score.

IC_j_	SDC			GPB			OL			Unit
	PHS	%	IC_j_ Score	PHS	%	IC_j_ Score	PHS	%	IC_j_ Score	
CC	FE	32	5.9 × 10^−1^ ± 9.7 × 10^−2 b^	DFLO	36	7.3 × 10^−1^ ± 1.3 × 10^−1 a^	DFLO	46	5.7 × 10^−1^ ± 4.1 × 10^−2 c^	kg CO_2e_
OD	Sd	60	1.2 × 10^−8^ ± 1.8 × 10^−9 a^	Sd	41	1.1 × 10^−8^ ± 1.8 × 10^−9 b^	Sd	47	6.6 × 10^−9^ ± 1.5 × 10^−8 c^	kg CFC11_e_
IR	Fert	51	1.2 × 10^−2^ ± 1.8 × 10^−3 b^	Fert	59	1.4 × 10^−2^ ± 2.3 × 10^−3 a^	Sd	32	6.6 × 10^−3^ ± 1.0 × 10^−2 c^	kBq ^235^U_e_
PhOF	DFLO	77	4.0 × 10^−3^ ± 6.4 × 10^−4 c^	DFLO	81	5.2 × 10^−3^ ± 8.8 × 10^−4 a^	DFLO	89	4.6 × 10^−3^ ± 8.8 × 10^−4 b^	kg NMVOC_e_
PM	DFLO	79	5.5 × 10^−8^ ± 8.9 × 10^−9 c^	DFLO	75	7.9 × 10^−8^ ± 1.3 × 10^−8 a^	DFLO	90	6.5 × 10^−8^ ± 1.1 × 10^−8 b^	disease inc.
A	DFLO	67	3.3 × 10^−3^ ± 5.2 × 10^−4 c^	DFLO	70	4.3 × 10^−3^ ± 7.2 × 10^−4 a^	DFLO	86	3.5 × 10^−3^ ± 7.8 × 10^−4 b^	mol H^+^_e_
FWE	Fert	52	3.5 × 10^−5^ ± 5.4 × 10^−6 c^	Fert	63	4.0 × 10^−5^ ± 6.5 × 10^−6 b^	TR	36	1.7 × 10^−5^ ± 2.1 × 10^−6 a^	kg P_e_
ME	DFLO	81	1.4 × 10^−3^ ± 2.2 × 10^−4 c^	DFLO	83	1.8 × 10^−3^ ± 3.1 × 10^−4 a^	DFLO	92	1.6 × 10^−3^ ± 3.0 × 10^−4 b^	kg N_e_
TE	DFLO	76	1.6 × 10^−2^ ± 2.5 × 10^−3 c^	DFLO	70	2.4 × 10^−2^ ± 4.0 × 10^−3 a^	DFLO	90	1.8 × 10^−2^ ± 3.2 × 10^−3 b^	mol N_e_
ETFW	Sd	90	8.2 × 10^1^ ± 1.3 × 10^1 a^	Sd	86	8.2 × 10^1^ ± 1.4 × 10^1 a^	Sd	96	1.4 × 10^2^ ± 6.0 × 10^2 b^	CTU_e_
C-HT	Sd	39	2.5 × 10^−10^ ± 3.9 × 10^−11 b^	TR	34	2.9 × 10^−10^ ± 4.7 × 10^−11 a^	TR	38	1.9 × 10^−10^ ± 2.7 × 10^−10 c^	CTU_h_
NC-HT	Sd	77	1.1 × 10^−8^ ± 1.6 × 10^−9 b^	Sd	75	1.3 × 10^−8^ ± 2.3 × 10^−9 a^	Sd	81	1.1 × 10^−8^ ± 4.0 × 10^−8 c^	CTU_h_
LU	FE	98	5.3 × 10^2^ ± 8.1 × 10^1 c^	FE	95	7.4 × 10^2^ ± 1.2 × 10^2 a^	FE	98	7.0 × 10^2^ ± 1.0 × 10^2 b^	Pt
WU	Sd	91	2.0 × 10^−1^ ± 3.2 × 10^−2 b^	FE	100	2.7 × 10^1^ ± 4.5 × 10^0 a^	DFLO	77	1.8 × 10^−2^ ± 4.4 × 10^−3 c^	m^3^ depriv.
RUF	DFLO	~100	2.6 × 10^0^ ± 4.3 × 10^−1 c^	DFLO	~100	3.6 × 10^0^ ± 6.2 × 10^−1 a^	DFLO	~100	3.5 × 10^0^ ± 4.7 × 10^−1 b^	MJ
RUMM	Fert	48	2.2 × 10^−6^ ± 3.4 × 10^−7 b^	Fert	60	2.4 × 10^−6^ ± 3.9 × 10^−7 a^	TR	39	9.7 × 10^−7^ ± 1.3 × 10^−6 c^	kg Sb_e_

In each row, values with the same letter have no significant difference at *p* < 0.05.

**Table 9 foods-13-00655-t009:** End-point environmental characterization of 1 kg of the dried pulses (SDC, GPB, and OL) at the farm gate using the PEF standard method: mean value and standard deviation of any normalized and weighted impact category (ICj) and corresponding percentage contribution, and Overall Weighted Sustainability Score (OWSS).

IC_j_	SDC		GPB		OL	
	[μPt]	[%]	[μPt]	[%]	[μPt]	[%]
CC	17.3 ± 3.0	16.7%	21.5 ± 4.0	6.1%	17.2 ± 18.4	14.1%
OD	0.015 ± 0.002	0.01%	0.014 ± 0.002	0.004%	0.01 ± 0.03	0.01%
IR	0.015 ± 0.002	0.1%	0.18 ± 0.03	0.1%	0.09 ± 0.20	0.1%
PhOF	5.0 ± 0.8	4.8%	6.5 ± 1.1	1.8%	5.8 ± 1.5	4.7%
PM	8.9 ± 1.5	8.6%	12.7 ± 2.2	3.6%	10.4 ± 2.3	8.5%
A	3.9 ± 0.6	3.8%	5.1 ± 0.9	1.4%	4.1 ± 1.3	3.4%
ME	2.2 ± 0.4	2.1%	2.9 ± 0.5	0.8%	2.6 ± 0.6	2.1%
FWE	0.6 ± 0.1	0.6%	0.7 ± 0.1	0.2%	0.3 ± 0.6	0.3%
TE	3.5 ± 0.6	3.4%	5.2 ± 0.9	1.5%	4.0 ± 0.9	3.2%
LU	54.1 ± 8.5	52.2%	75.9 ± 12.7	21.5%	71.9 ± 12.2	58.6%
WU	1.6 ± 0.3	1.5%	214.7 ± 36.4	60.8%	0.14 ± 0.05	0.1%
RUF	3.6 ± 0.6	3.5%	4.9 ± 0.9	1.4%	4.8± 0.6	3.9%
RUMM	2.7 ± 0.4	2.6%	3.0 ± 0.5	0.8%	1.3 ± 2.5	1.1%
OWSS	103.6 ± 16.7	100.0%	353.3 ± 60.2	100.0%	122.6 ± 41.1	100.0%

**Table 10 foods-13-00655-t010:** Environmental profile for 1 kg of dried pulses (SDC, GPB, and OL) in cradle-to-grave perspective using the PEF standard method: primary (PHS) and secondary (SHS) hotspots with relative percentage contributions, mean values, and standard deviations across mid-point impact categories (Icj).

IC_j_	SDC					GPB					OL					Unit
	PHS(%)	SHS(%)	IC_j_ Score			PHS(%)	SHS(%)	IC_j_ Score			PHS(%)	SHS(%)	IC_j_ Score			
CC	CU (29)	FPh (28)	2.5	±	1.3 × 10^−1^	FPh (33)	CU (25)	2.6	±	1.7 × 10^−1^	FPh (36)	CU (20)	2.6	±	4.5	kg CO_2e_
OD	CU (44)	FPh (24)	5.7 × 10^−8^	±	2.3 × 10^−9^	CU (41)	FPh (23)	5.3 × 10^−8^	±	2.4 × 10^−9^	CU (37)	TR (21)	4.7 × 10^−8^	±	1.6 × 10^−7^	kg CFC11_e_
IR	CU (33)	PR (25)	1.4 × 10^−1^	±	2.4 × 10^−3^	CU (31)	PR (24)	1.4 × 10^−1^	±	3.2 × 10^−3^	CU (30)	PR (27)	1.3 × 10^−1^	±	1.1 × 10^−1^	kBq ^235^U_e_
PhOF	FPh (42)	TR (27)	1.1 × 10^−2^	±	8.2 × 10^−4^	FPh (49)	TR (25)	1.3 × 10^−2^	±	1.2 × 10^−3^	FPh (55)	TR (23)	1.4 × 10^−2^	±	7.1 × 10^−3^	kg NMVOC_e_
PM	FPh (47)	TR (30)	1.4 × 10^−7^	±	1.1 × 10^−8^	FPh (56)	TR (26)	1.7 × 10^−7^	±	1.8 × 10^−8^	FPh (60)	TR (24)	1.8 × 10^−7^	±	8.1 × 10^−8^	disease inc.
A	FPh (38)	TR (20)	1.0 × 10^−2^	±	6.7 × 10^−4^	FPh (44)	TR (19)	1.2 × 10^−2^	±	9.8 × 10^−4^	FPh (48)	TR (18)	1.2 × 10^−2^	±	7.1 × 10^−3^	mol H^+^_e_
I	PMP (28)	CU (22)	3.9 × 10^−4^	±	8.1 × 10^−6^	PMP (28)	CU (20)	3.9 × 10^−4^	±	9.8 × 10^−6^	PMP (30)	CU (19)	3.7 × 10^−4^	±	2.3 × 10^−4^	kg P_e_
ME	WD (38)	FPh (32)	5.2 × 10^−3^	±	3.3 × 10^−4^	FPh (39)	WD (35)	5.7 × 10^−3^	±	4.4 × 10^−4^	FPh (42)	TR (36)	6.6 × 10^−3^	±	2.3 × 10^−3^	kg N_e_
TE	FPh (54)	TR (22)	3.6 × 10^−2^	±	3.3 × 10^−3^	FPh (64)	TR (18)	4.6 × 10^−2^	±	5.4 × 10^−3^	FPh (65)	TR (17)	4.7 × 10^−2^	±	2.5 × 10^−2^	mol N_e_
ETFW	FPh (90)	WD (4)	1.2 × 10^2^	±	1.6 × 10^1^	FPh (90)	WD (4)	1.2 × 10^2^	±	2.0 × 10^1^	FPh (95)	WD (2)	3.3 × 10^2^	±	6.6 × 10^3^	CTU_e_
C-HT	TR (47)	CU (18)	1.4 × 10^−9^	±	5.0 × 10^−11^	TR (48)	FPh (17)	1.4 × 10^−9^	±	6.5 × 10^−11^	TR (50)	FPh (16)	1.4 × 10^−9^	±	2.9 × 10^−9^	CTU_h_
NC-HT	FPh (37)	PMP (25)	3.5 × 10^−8^	±	2.1 × 10^−9^	FPh (42)	PMP (23)	3.8 × 10^−8^	±	3.1 × 10^−9^	FPh (44)	PMP (23)	4.5 × 10^−8^	±	4.5 × 10^−7^	CTU_h_
LU	FPh (97)	PMP (2)	7.1 × 10^2^	±	1.0 × 10^2^	FPh (98)	PMP (2)	9.9 × 10^2^	±	1.6 × 10^2^	FPh (98)	PMP (1)	1.3 × 10^3^	±	6.1 × 10^2^	Pt
WU	FPh (37)	CU (24)	7.0 × 10^−1^	±	4.1 × 10^−2^	FPh (99)	CU (0.5)	3.6 × 10^1^	±	6.0	CU (33)	PR (32)	4.4 × 10^−1^	±	4.2 × 10^−2^	m^3^ depriv.
RUF	FPh (76)	PMP (21)	4.5	±	5.5 × 10^−1^	FPh (81)	PMP (16)	5.7	±	8.4 × 10^−1^	FPh (85)	PMP (13)	7.2	±	8.3 × 10^−1^	MJ
RUMM	PMP (44)	CU (22)	1.8 × 10^−5^	±	5.5 × 10^−7^	PMP (44)	CU (21)	1.9 × 10^−5^	±	6.3 × 10^−7^	PMP (48)	TR (21)	1.7 × 10^−5^	±	1.4 × 10^−5^	kg Sb_e_

**Table 11 foods-13-00655-t011:** End-point business-to-business environmental characterization for 1 kg of dried pulses (SDC, GPB, and OL) according to the PEF standard method: mean value and standard deviation of normalized and weighted impact categories (IC_j_) and corresponding percentage contributions.

IC_j_	SDC		GPB		OL	
	[μPt]	[%]	[μPt]	[%]	[μPt]	[%]
CC	72.4 ± 3.8	28.5	75.9 ± 5.2	13.1	75.5 ± 131.0	22.9
OD	0.073 ± 0.003	0.03	0.1 ± 0.0	0.01	0.1 ± 0.2	0.02
IR	1.8 ± 0.0	0.7	1.8 ± 0.0	0.3	1.7 ± 1.4	0.5
PhOF	14.0 ± 1.0	5.5	15.9 ± 1.5	2.7	17.4 ± 8.8	5.3
PM	22.2 ± 1.8	8.7	27.4 ± 2.9	4.7	28.9 ± 13.0	8.8
A	12.5 ± 0.8	4.9	13.9 ± 1.2	2.4	14.4 ± 8.5	4.4
ME	8.2 ± 0.5	3.2	9.1 ± 0.7	1.6	10.5 ± 3.7	3.2
FWE	7.2 ± 0.2	2.8	7.2 ± 0.2	1.2	6.7 ± 4.3	2.0
TE	8.0 ± 0.7	3.1	10.2 ± 1.2	1.8	10.3 ± 5.5	3.1
LU	72.3 ± 10.7	28.5	102.1 ± 16.6	17.6	128.9 ± 62.4	39.2
WU	5.5 ± 0.3	2.2	286.0 ± 47.5	49.2	3.5 ± 0.3	1.1
RUF	6.1 ± 0.8	2.4	7.9 ± 1.1	1.4	9.8 ± 1.1	3.0
RUMM	23.4 ± 0.7	9.2	23.5 ± 0.8	4.0	21.4 ± 17.8	6.5
OWSS	253.7 ± 21.4	100.0	581.0 ± 79.0	100.0	328.9 ± 258.0	100.0

**Table 12 foods-13-00655-t012:** End-point business-to-consumer environmental characterization and Overall Weighted Sustainability Score (OWSS) for 1 kg of dried pulses (SDC, GPB, and OL) according to the PEF standard method, using in situ cultivated seeding material: mean value and standard deviation of overall (ICj) and normalized and weighted (NWICj) impact categories, and OWSS.

IC_j_	SDC						GPB						OL					
	ICj Score ^#^			NWIC_j_ [μPt]			ICj Score ^#^			NWIC_j_ [μPt]			ICj Score ^#^			NWIC_j_ [μPt]		
CC	2.4 × 10^0^ ± 1.4 × 10^−1^			71.9	±	4.1	2.6 × 10^0^ ± 2.1 × 10^−1^			76.8	±	6.1	2.5 × 10^0^ ± 1.6 × 10^−1^			73.0	±	4.7
OD	4.8 × 10^−8^ ± 1.2 × 10^−9^			0.06	±	0.00	4.8 × 10^−8^ ± 1.8 × 10^−9^			0.06	±	0.00	4.0 × 10^−8^ ± 1.1 × 10^−9^			0.05	±	0.00
IR	1.4 × 10^−1^ ± 2.3 × 10^−3^			1.79	±	0.03	1.4 × 10^−1^ ± 3.4 × 10^−3^			1.81	±	0.04	1.3 × 10^−1^ ± 1.3 × 10^−3^			1.64	±	0.02
PhOF	1.1 × 10^−2^ ± 9.5 × 10^−4^			14.2	±	1.2	1.3 × 10^−2^± 1.5 × 10^−3^			16.6	±	1.8	1.5 × 10^−2^ ± 1.4 × 10^−3^			18.5	±	1.7
PM	1.4 × 10^−7^ ± 1.3 × 10^−8^			22.7	±	2.1	1.7 × 10^−7^ ± 2.0 × 10^−8^			27.1	±	3.3	1.9 × 10^−7^ ± 1.9 × 10^−8^			31.0	±	3.1
A	1.0 × 10^−2^ ± 7.6 × 10^−4^			12.5	±	0.9	1.2 × 10^−2^ ± 1.2 × 10^−3^			14.3	±	1.4	1.3 × 10^−2^ ± 1.0 × 10^−3^			15.1	±	1.2
ME	5.2 × 10^−3^ ± 3.6 × 10^−4^			8.3	±	0.6	5.9 × 10^−3^ ± 5.3 × 10^−4^			9.4	±	0.8	6.9 × 10^−3^ ± 5.2 × 10^−4^			11.0	±	0.8
FWE	3.8 × 10^−4^ ± 7.8 × 10^−6^			7.1	±	0.1	3.9 × 10^−4^ ± 1.1 × 10^−5^			7.2	±	0.2	3.6 × 10^−4^ ± 6.0 × 10^−6^			6.6	±	0.1
TE	3.7 × 10^−2^± 3.8 × 10^−3^			8.2	±	0.8	4.5 × 10^−2^ ± 5.8 × 10^−3^			9.9	±	1.3	5.0 × 10^−2^ ± 5.3 × 10^−3^			11.1	±	1.2
ETFW	2.3 × 10^1^ ± 2.1 × 10^0^			0.0	±	0.0	2.7 × 10^1^ ± 3.3 × 10^0^			0.0	±	0.0	2.2 × 10^1^ ± 1.6 × 10^0^			0.0	±	0.0
C-HT	1.3 × 10^−9^ ± 3.7 × 10^−11^			0.0	±	0.0	1.3 × 10^−9^ ± 5.5 × 10^−11^			0.0	±	0.0	1.2 × 10^−9^ ± 3.6 × 10^−11^			0.0	±	0.0
NC-HT	2.5 × 10^−8^ ± 6.8 × 10^−10^			0.0	±	0.0	2.6 × 10^−8^ ± 1.0 × 10^−9^			0.0	±	0.0	2.4 × 10^−8^ ± 6.6 × 10^−10^			0.0	±	0.0
LU	7.7 × 10^2^ ± 1.3 × 10^2^			79.4	±	13.2	1.1 × 10^3^ ± 2.0 × 10^2^			109.2	±	20.4	1.4 × 10^3^ ± 2.0 × 10^2^			145.4	±	20.3
WU	4.6 × 10^−1^ ± 4.7 × 10^−3^			3.61	±	0.04	4.1 × 10^1^ ± 7.8 × 10^0^			320.15	±	61.24	4.4 × 10^− 1^ ± 5.1 × 10^−3^			3.47	±	0.04
RUF	4.9 × 10^0^ ± 6.9 × 10^−1^			6.7	±	0.9	6.3 × 10^0^ ± 1.1 × 10^0^			8.7	±	1.4	8.1 × 10^0^ ± 1.1 × 10^0^			11.1	±	1.5
RUMM	1.8 × 10^−5^ ± 4.9 × 10^−7^			22.6	±	0.6	1.8 × 10^−5^ ± 6.8 × 10^−7^			23.2	±	0.9	1.6 × 10^−5^ ± 4.0 × 10^−7^			20.8	±	0.5
OWSS				259	±	25				624	±	99				349	±	35

**^#^** The unit of each j-th impact category score is given in the Nomenclature section.

**Table 13 foods-13-00655-t013:** End-point business-to-consumer environmental characterization and Overall Weighted Sustainability Score (OWSS) for 1 kg of dried malted and decorticated pulses (MDSDC, MDGPB, and MDOL) according to the PEF standard method, using in situ cultivated seeding material: mean value and standard deviation of overall (ICj) and normalized and weighted (NWICj) impact categories, and OWSS.

IC_j_	MDSDC						MDGPB						MDOL					
	ICj Score ^#^			NWIC_j_ [μPt]			ICj Score ^#^			NWIC_j_ [μPt]			ICj Score ^#^			NWIC_j_ [μPt]		
CC	2.8 × 10^0^ ± 1.7 × 10^−1^			82.2	±	5.0	3.0 × 10^0^ ± 2.3 × 10^−1^			89.6	±	6.9	3.0 × 10^0^ ± 1.9 × 10^−1^			88.2	±	5.4
OD	5.3 × 10^−8^ ± 1.5 × 10^−9^			0.07	±	0.00	5.4 × 10^−8^ ± 2.1 × 10^−9^			0.070	±	0.003	5.0 × 10^−8^ ± 1.4 × 10^−9^			0.06	±	0.00
IR	1.9 × 10^−1^ ± 3.6 × 10^−3^			2.36	±	0.05	1.9 × 10^−1^± 4.7 × 10^−3^			2.41	±	0.06	1.8 × 10^−1^ ± 2.9 × 10^−3^			2.27	±	0.04
PhOF	1.3 × 10^−2^ ± 1.2 × 10^−3^			16.5	±	1.4	1.5 × 10^−2^ ± 1.7 × 10^−3^			19.3	±	2.1	1.7 × 10^−2^ ± 1.6 × 10^−3^			21.8	±	2.0
PM	1.6 × 10^−7^ ± 1.6 × 10^−8^			25.7	±	2.6	1.9 × 10^−7^ ± 2.3 × 10^−8^			30.9	±	3.7	2.2 × 10^−7^ ± 2.2 × 10^−8^			35.6	±	3.6
A	1.2 × 10^−2^ ± 9.3 × 10^−4^			14.8	±	1.1	1.4 × 10^−2^± 1.3 × 10^−3^			17.0	±	1.6	1.5 × 10^−2^ ± 1.2 × 10^−3^			18.1	±	1.4
ME	5.8 × 10^−3^ ± 4.3 × 10^−4^			9.3	±	0.7	6.6 × 10^−3^ ± 6.0 × 10^−4^			10.6	±	1.0	7.8 × 10^−3^ ± 5.9 × 10^−4^			12.5	±	0.9
FWE	4.4 × 10^−4^ ± 9.5 × 10^−6^			8.2	±	0.2	4.6 × 10^−4^ ± 1.3 × 10^−5^			8.4	±	0.2	4.3 × 10^−4^ ± 7.1 × 10^−6^			8.0	±	0.1
TE	4.3 × 10^−2^ ± 4.6 × 10^−3^			9.5	±	1.0	5.2 × 10^−2^ ± 6.7 × 10^−3^			11.6	±	1.5	5.9 × 10^−2^ ± 6.1 × 10^−3^			13.0	±	1.4
ETFW	2.6 × 10^1^ ± 2.5			0.0	±	0.0	3.1 × 10^1^± 3.7 × 10^0^			0.0	±	0.0	2.5 × 10^1^ ± 1.8 × 10^0^			0.0	±	0.0
C-HT	1.4 × 10^−9^ ± 4.5 × 10^−11^			0.0	±	0.0	1.4 × 10^−9^ ± 6.3 × 10^−11^			0.0	±	0.0	1.4 × 10^−9^ ± 4.1 × 10^−11^			0.0	±	0.0
NC-HT	2.7 × 10^−8^ ± 8.2 × 10^−10^			0.0	±	0.0	2.9 × 10^−8^ ± 1.1 × 10^−9^			0.0	±	0.0	2.8 × 10^−8^ ± 7.4 × 10^−10^			0.0	±	0.0
LU	9.1 × 10^2^ ± 1.5 × 10^2^			93.3	±	15.8	1.3 × 10^3^ ± 2.3 × 10^2^			128.5	±	23.3	1.7 × 10^3^ ± 2.3 × 10^2^			170.5	±	23.1
WU	6.4 × 10^−1^ ± 1.1 × 10^−2^			5.03	±	0.09	4.8 × 10^1^ ± 8.9 × 10^0^			378.38	±	70.11	6.4 × 10^−1^ ± 1.2 × 10^−2^			5.02	±	0.09
RUF	5.5 ± 8.4 × 10^−1^			7.6	±	1.1	7.3 × 10^0^ ± 1.2 × 10^0^			10.0	±	1.7	9.3 × 10^0^ ± 1.2 × 10^0^			12.8	±	1.7
RUMM	1.8 × 10^−5^ ± 5.6 × 10^−7^			23.3	±	0.7	1.9 × 10^−5^ ± 7.1 × 10^−7^			24.3	±	0.9	1.7 × 10^−5^ ± 4.1 × 10^−7^			21.9	±	0.5
OWSS				298	±	30				731	±	113				410	±	40

**^#^** The unit of each j-th impact category score is given in the Nomenclature section.

## Data Availability

The original contributions presented in the study are included in the article and Appendix A, further inquiries can be directed to the corresponding author.

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
