# Peer review of "A Comprehensive Study from Cradle-to-Grave on the Environmental Profile of Malted Legumes"

_foods, 2024, doi:10.3390/foods13050655_

Round 1

Reviewer 1 Report

Comments and Suggestions for Authors

This issue is interesting. However, there remains room for improvement in terms of both logical structure and content. To enhance the overall quality and credibility of the conclusions drawn, it is crucial to provide a more detailed and transparent explanation of the methods and empirical processes employed.

Specifically, the methods utilized in the study should be elucidated comprehensively to ensure readers have a clear understanding of the research approach. This transparency not only fosters credibility but also facilitates the replication of the study, a fundamental aspect of scientific research.

Furthermore, it is advisable to bolster the discussion section, particularly regarding the methodology. Expanding upon the methodological choices made, the rationale behind them, and their implications can provide valuable insights. This not only aids in contextualizing the research but also allows for a deeper exploration of the study's strengths and limitations.

To enhance the clarity and effectiveness of writing the methods and empirical results sections for the article "A comprehensive study from cradle-to-grave on the environmental profile of malted legumes," consider the following detailed suggestions:

  1. Introduction to the Study:
    • Begin with a brief overview of the importance of studying the environmental profile of malted legumes.
    • Highlight the significance of reducing anti-nutrient content in pulses through malting to increase consumption.
    • Emphasize the relevance of Life Cycle Analysis (LCA) in assessing the environmental impact of food production.
  2. Description of Sample Pulses:
    • Provide detailed descriptions of Solco Dritto chickpeas (SDC), Gradoli Purgatory beans (GPB), and Onano lentils (OL), including their nutritional value, traditional uses, and regional significance.
    • Explain the rationale behind selecting these specific pulses for the study.
  3. Malting Process:
    • Describe the malting process used for each type of pulse, including soaking, germination, and drying stages.
    • Mention any specific conditions or parameters controlled during malting, such as temperature, humidity, and duration.
  4. Life Cycle Analysis (LCA):
    • Explain the methodology and scope of the LCA conducted for assessing the environmental impact of malted legumes.
    • Detail the life cycle stages considered, such as agricultural production, processing, transportation, and consumption.
    • Clarify the system boundaries and functional unit used for the analysis.
  5. Environmental Impact Assessment:
    • Present the findings of the LCA in a structured manner, focusing on key impact categories such as carbon footprint, land use, and water use.
    • Use tables, graphs, or charts to illustrate the environmental impact data for each type of malted legume.
    • Discuss the significance of each impact category and its contribution to the overall environmental profile.
  6. Mitigation Strategies:
    • Propose mitigation strategies based on the identified environmental impacts, addressing issues related to land use, water consumption, and climate change.
    • Provide specific recommendations for optimizing energy consumption during malting, transitioning to renewable energy sources, and improving transportation efficiency.
  7. Implications for Food Product Development:
    • Discuss the potential applications of malted legumes in gluten-free, low-fat, and phytate-specific food products for individuals with dietary restrictions.
    • Highlight the role of sustainable practices in promoting innovation and improving dietary choices.
  8. Conclusion:
    • Summarize the main findings of the study, emphasizing the importance of addressing environmental concerns in pulse production and consumption.
    • Reinforce the potential benefits of adopting sustainable practices in the food industry and the importance of further research in this area.

The other suggestions are:

  1. Introduction:
    • Provide a brief overview of the importance of studying the environmental profile of malted legumes.
    • Highlight the significance of reducing anti-nutrient content in pulses through malting to increase consumption.
    • Emphasize the relevance of Life Cycle Analysis (LCA) in assessing the environmental impact of food production.
  2. Description of Sample Pulses:
    • Provide detailed descriptions of Solco Dritto chickpeas (SDC), Gradoli Purgatory beans (GPB), and Onano lentils (OL), including their nutritional value, traditional uses, and regional significance.
    • Explain the rationale behind selecting these specific pulses for the study.
  3. Malting Process:
    • Describe the malting process used for each type of pulse, including soaking, germination, and drying stages.
    • Mention any specific conditions or parameters controlled during malting, such as temperature, humidity, and duration.
  4. Life Cycle Analysis (LCA):
    • Explain the methodology and scope of the LCA conducted for assessing the environmental impact of malted legumes.
    • Detail the life cycle stages considered, such as agricultural production, processing, transportation, and consumption.
    • Clarify the system boundaries and functional unit used for the analysis.
  5. Environmental Impact Assessment:
    • Present the findings of the LCA in a structured manner, focusing on key impact categories such as carbon footprint, land use, and water use.
    • Use tables, graphs, or charts to illustrate the environmental impact data for each type of malted legume.
    • Discuss the significance of each impact category and its contribution to the overall environmental profile.
  6. Mitigation Strategies:
    • Propose mitigation strategies based on the identified environmental impacts, addressing issues related to land use, water consumption, and climate change.
    • Provide specific recommendations for optimizing energy consumption during malting, transitioning to renewable energy sources, and improving transportation efficiency.
  7. Implications for Food Product Development:
    • Discuss the potential applications of malted legumes in gluten-free, low-fat, and phytate-specific food products for individuals with dietary restrictions.
    • Highlight the role of sustainable practices in promoting innovation and improving dietary choices.
  8. Conclusion:
    • Summarize the main findings of the study, emphasizing the importance of addressing environmental concerns in pulse production and consumption.
    • Reinforce the potential benefits of adopting sustainable practices in the food industry and the importance of further research in this area.

Comments on the Quality of English Language

Pls proofreading.

Author Response

We thank referee#1 for his/her suggestions, even though we wonder if the reviewer has effectively read the paper. Moreover, his/her suggestions were duplicated, as marked in yellow above.

We shall try to answer to his/her specific comments as reported below.

Ref#1:   This issue is interesting. However, there remains room for improvement in terms of both logical structure and content. To enhance the overall quality and credibility of the conclusions drawn, it is crucial to provide a more detailed and transparent explanation of the methods and empirical processes employed.

Specifically, the methods utilized in the study should be elucidated comprehensively to ensure readers have a clear understanding of the research approach. This transparency not only fosters credibility but also facilitates the replication of the study, a fundamental aspect of scientific research.

Furthermore, it is advisable to bolster the discussion section, particularly regarding the methodology. Expanding upon the methodological choices made, the rationale behind them, and their implications can provide valuable insights. This not only aids in contextualizing the research but also allows for a deeper exploration of the study's strengths and limitations.

The paper was structured in accordance with ISO Standards 14040 and 14044, following the typical four stages: Goal and scope definition (§3.1), inventory analysis (§3.2), impact assessment (§3.3), and interpretation of results (§4.). This logical framework mirrors the structure utilized in numerous previous papers of ours, such as.

Cimini A, Moresi M (2016) Carbon footprint of a pale lager packed in different formats: assessment and sensitivity analysis based on transparent data. Journal of Cleaner Production 112: 4196-4213.

Cimini A, Moresi M (2017) Effect of brewery size on the main process parameters and cradle-to- grave carbon footprint of lager beer. Journal of Industrial Ecology, 22 (5), 1139- 1155 (DOI: 10.1111/jiec.12642)

Cimini A, Cibelli M, Moresi M (2019) Cradle-to-grave carbon footprint of dried organic pasta: assessment and potential mitigation measures. J Sci Food Agric, 99, 5303–5318 .

Cimini A, Cibelli M, Moresi M (2020) Environmental impact of pasta. In Galanakis C (Ed.) Environmental Impact of Agro-Food Industry and Food Consumption. Chp. 5. Academic Press, San Diego, CA, USA, pp. 101-127. https://doi.org/10.1016/B978-0-12-821363-6.00005-9

Cibelli M, Cimini A, Cerchiara G, Moresi M (2021) Carbon Footprint of different methods of coffee preparation. Sustainable Production and Consumption, 27: 1614-1625. https://doi.org/10.1016/j.spc.2021.04.004

Cimini A, Moresi M (2022). Environmental impact of the main household cooking systems - a survey. Italian Journal of Food Science, 34 (1): 86–113; doi 10.15586/ijfs.v34i1.2170

Falciano A, Cimini A, Masi P, Moresi M (2022) Carbon Footprint of a Typical Neapolitan Pizzeria. Sustainability, 14(5), 3125; https://doi.org/10.3390/su14053125

Cimini A, Sestili F, Moresi M (2022) Environmental profile of a novel high-amylose bread wheat fresh pasta with low glycemic index. Foods 2022, 11, 3199. https://doi.org/10.3390/foods11203199

Cimini A, Moresi M (2023) Fresh versus dry pasta: what is the difference in their environmental impact? Chemical Engineering Transactions, 102: 7-12. (DOI: 10.3303/CET23102002). https://www.cetjournal.it/cet/23/102/002.pdf

Regarding §3.1, the functional unit selected coincided with a PP bag containing 500 g of dried pulses usually commercialized in supermarkets and retailers in the Latium region, as shown in the following pictures and described at §3.2.4.

SDC

GPB

OL

As for the referee’s statement pointing out that transparency not only fosters credibility but also facilitates the replication of the study, a fundamental aspect of scientific research, we emphasize that in our opinion this paper might be the most transparent one ever published so far. Not only are all the different phases of the life cycle of dried pulses accurately described (see subsections §3.2.1 to §32.2.10), but also all the material and energy balances are reported in the Supplementary Material 2. Furthermore, all stages of the dry pulse network's product life cycle are thoroughly outlined in Supplementary Material 1, enabling any interested reader to replicate our study using the LCA model developed in SimaPro software.

  1. Introduction to the Study:
    • Begin with a brief overview of the importance of studying the environmental profile of malted legumes.
    • Highlight the significance of reducing anti-nutrient content in pulses through malting to increase consumption.
    • Emphasize the relevance of Life Cycle Analysis (LCA) in assessing the environmental impact of food production.

The Introduction section was structured as suggested by the referee (see Lines 53-96).

  1. Description of Sample Pulses:
    • Provide detailed descriptions of Solco Dritto chickpeas (SDC), Gradoli Purgatory beans (GPB), and Onano lentils (OL), including their nutritional value, traditional uses, and regional significance.
    • Explain the rationale behind selecting these specific pulses for the study.
  2. Malting Process:
    • Describe the malting process used for each type of pulse, including soaking, germination, and drying stages.
    • Mention any specific conditions or parameters controlled during malting, such as temperature, humidity, and duration.

A new section 2 entitled Pulses: cultivation and utilization, market prospects, and ecological implications was added by reporting the basic composition of the pulses in the original state and malted form (new Table 1).

The use of malted and decorticated pulse flours as ingredient for the formulation of fresh egg-pasta was documented in reference [19] for OLs, and in reference [33] for SDCs and GPBs. The composition, antinutrient contents and in vitro glycemic index of the resulting fresh egg-pastas were also reported. Finally, the malting process used was described at section §3.2.10.

  1. Life Cycle Analysis (LCA):
    • Explain the methodology and scope of the LCA conducted for assessing the environmental impact of malted legumes.
    • Detail the life cycle stages considered, such as agricultural production, processing, transportation, and consumption.
    • Clarify the system boundaries and functional unit used for the analysis.

The methodology used was described at new section 3 by referring to the 4 stages of the LCA tool.

  1. Environmental Impact Assessment:
    • Present the findings of the LCA in a structured manner, focusing on key impact categories such as carbon footprint, land use, and water use.
    • Use tables, graphs, or charts to illustrate the environmental impact data for each type of malted legume.
    • Discuss the significance of each impact category and its contribution to the overall environmental profile.

The interpretation of the LCA results was divided into distinct sections, initially focusing on the environmental profile of harvested pulse seeds at the farm gate (§4.1). Subsequently, the cradle-to-grave environmental profile of dry pulses (§4.2) and dry malted and decorticated pulses (§4.3) was presented and discussed. The characterization of each pulse product was illustrated using tables and column diagrams.

  1. Mitigation Strategies:
    • Propose mitigation strategies based on the identified environmental impacts, addressing issues related to land use, water consumption, and climate change.
    • Provide specific recommendations for optimizing energy consumption during malting, transitioning to renewable energy sources, and improving transportation efficiency.

Section 4.4 delved into suggested options aimed at enhancing the sustainability of dry pulses in their original form as well as malted and decorticated states, addressing issues concerning the mitigation of Land use (§4.4.1), Water use (§4.4.2), and Climate change (§4.4.3).

  1. Implications for Food Product Development:
    • Discuss the potential applications of malted legumes in gluten-free, low-fat, and phytate-specific food products for individuals with dietary restrictions.
    • Highlight the role of sustainable practices in promoting innovation and improving dietary choices.

The potential application of malted legumes was outlined in previous papers [18-19, 33].

  1. Conclusion:
    • Summarize the main findings of the study, emphasizing the importance of addressing environmental concerns in pulse production and consumption.
    • Reinforce the potential benefits of adopting sustainable practices in the food industry and the importance of further research in this area.

The main findings of the study were summarized to raise awareness among the general consumers regarding the heightened environmental impact associated with functional foods, such as malted and decorticated pulses. Furthermore, the necessity for additional research to evaluate consumer acceptability and willingness to pay for enhanced dietary choices was emphasized (see Lines 954-964).

Reviewer 2 Report

Comments and Suggestions for Authors

The paper presents a comprehensive study assessing the environmental profile of three malted legumes - Solco Dritto chickpeas (SDC), Gradoli Purgatory beans (GPB), and Onano lentils (OL) - from the Latium region of Italy. The malting process aimed to reduce anti-nutrient content, making the pulses more suitable for consumption. Using Life Cycle Analysis (LCA), the environmental impact of these malted legumes was evaluated, revealing carbon footprints ranging from 2.8 to 3.0 kg CO2e per kg of malted legumes. The Overall Weighted Sustainability scores (OWSS) ranged from 298±30 to 410±40 or 731±113 µPt for different malted legumes, showing a 13% to 17% increase compared to untreated dry seeds. Land use impact was a significant factor contributing to OWSS, with water use emerging as a crucial factor in GPB cultivation. The paper discusses mitigation strategies to improve sustainability, such as selecting drought-tolerant bean varieties and optimizing electricity consumption during malting.

My Comments:

1) Can you provide more details on the specific malting process used for each type of legume? How does this process affect the nutritional composition and digestibility of the legumes?

2) The paper mentions the use of the Product Environmental Footprint standard method. Could you elaborate on how this method was applied in the study and its advantages over other LCA methodologies?

3) In the methodology section, the choice of functional unit (500 g of legumes) is mentioned. How was this quantity determined, and how does it impact the overall findings of the study?

4) The study discusses the impact of land use and water use on the environmental footprint. Could you explain how these factors were quantified and whether any local or regional variations were considered?

5) Regarding the mitigation strategies proposed, what are the potential challenges or limitations in implementing these strategies at a larger scale, especially concerning agricultural practices and consumer behavior?

6) The paper highlights the role of crop yield in reducing environmental impact. Can you discuss any ongoing efforts or research initiatives aimed at improving crop yield for the studied legumes?

7) The study suggests transitioning to photovoltaic electricity and upgrading transport vehicles. What are the economic implications of these changes, and how feasible are they for small-scale producers?

8) Regarding the assessment of cooking energy consumption, are there any additional factors or variables that were considered, such as variations in cooking methods or fuel sources?

9) The paper mentions the potential use of malted and decorticated pulses in specific food products for individuals with dietary restrictions. Have there been any consumer studies or market analyses conducted to assess the demand for such products?

10) Finally, in the conclusions section, the paper discusses the trade-off between reduced cooking times and increased carbon footprint. Can you provide further insights into how this trade-off was evaluated and its implications for consumer choices and environmental sustainability?

Author Response

1) Can you provide more details on the specific malting process used for each type of legume? How does this process affect the nutritional composition and digestibility of the legumes?

As also suggested by referee#1, a new section 2 entitled Pulses: cultivation and utilization, market prospects, and ecological implications was added by reporting the basic composition of the pulses in the original state and malted form (new Table 1). Moreover, the malting process used was described at section §3.2.10.

2) The paper mentions the use of the Product Environmental Footprint standard method. Could you elaborate on how this method was applied in the study and its advantages over other LCA methodologies?

The standard method used in this work was applied as recommended by EU Commission [33].

3) In the methodology section, the choice of functional unit (500 g of legumes) is mentioned. How was this quantity determined, and how does it impact the overall findings of the study?

The functional unit selected coincided with the bags usually commercialized in supermarkets and retailers in the Latium region, as shown in the following pictures and described at §3.2.4.

SDC

GPB

OL

4) The study discusses the impact of land use and water use on the environmental footprint. Could you explain how these factors were quantified and whether any local or regional variations were considered?

The categories of land use (LU) and water use (WU) were calculated as specified by the PEF method. The score of LU was estimated using the LANCA® v 2.2 baseline model (Bos et al. (2016)., while that of WU via the Available Water Remaining (AWARE) model (Bouley et al., 2018). The characterized impact score of any of these impact categories were then normalized with respect to the impacts caused by one person living in the world for one year, these being equal to 1.33x106 Pt and 1.15x104 m3e of deprived water, respectively. No regional data were utilized; instead, the irrigation water for cultivating GPBs originated from the consortium of the river Paglia and was categorized as a natural input in the SimaPro software under the characterization of Water, river, IT.

Bos U, Horn R, Beck T, Lindner JP, Fischer M (2016) LANCA® - Characterisation factors for Life Cycle Impact Assessment, Version 2.0. Fraunhofer Verlag, Stuttgart, DE.

Boulay A-M, Bare J, Benini L, Berger M, Lathuillière MJ, Manzardo A, Margni M, Motoshita M, Núñez M, Valerie-Pastor A, Ridoutt B, Oki T, Worbe S, Pfister S (2018) The WULCA consensus characterization model for water scarcity footprints: assessing impacts of water consumption based on available water remaining (AWARE). International Journal of Life Cycle Assessment 23(2): 368–378.

5) Regarding the mitigation strategies proposed, what are the potential challenges or limitations in implementing these strategies at a larger scale, especially concerning agricultural practices and consumer behavior?

The mitigation strategies outlined in this paper stem from an assessment of the primary hotspots associated with the pulses, whether in their original or malted state. Notably, these strategies are tailored to the small-scale production context, characteristic of the niche products examined here. Specifically, the feasibility of incorporating solar energy is closely linked to the recent installation of photovoltaic panels on a portion of the warehouse roofs at the target factory mentioned in this study, resulting in the recovery of up to 19,000 kWh/year. Plans are underway to further expand the paneling surfaces to enhance solar energy utilization (refer to Lines 896-900).

6) The paper highlights the role of crop yield in reducing environmental impact. Can you discuss any ongoing efforts or research initiatives aimed at improving crop yield for the studied legumes?

Currently, there are no research initiatives aimed at improving crop yields for the three legumes examined here that are engaging the agricultural cooperatives in the specific production area. Consequently, the crop yields attained for SDCs and OLs in a neighboring area were reported, along with their impact on the Overall Weighted Sustainability Score (OWSS) (refer to §4.4.1).

7) The study suggests transitioning to photovoltaic electricity and upgrading transport vehicles. What are the economic implications of these changes, and how feasible are they for small-scale producers?

The economic feasibility of transitioning to photovoltaic electricity and upgrading transportation vehicles was beyond the scope of this paper. However, it may appear feasible, even for small-scale producers like the target factory examined in this study, although their successful implementation depends on the specific economic circumstances of the entities involved. Please refer to the revised comments at Lines 916-925 for further details.

8) Regarding the assessment of cooking energy consumption, are there any additional factors or variables that were considered, such as variations in cooking methods or fuel sources?

As detailed in Appendix 1, the energy consumption for cooking the three pulses, whether in their original state or malted, utilizing a Liquified Petroleum Gas (LPG) cooking stove, an electric plate, or an induction hob, is presented in Table A3.

9) The paper mentions the potential use of in specific food products for individuals with dietary restrictions. Have there been any consumer studies or market analyses conducted to assess the demand for such products?

The utilization of malted and decorticated pulse flours as ingredients for the formulation of fresh egg-pasta was documented in reference [19] for OLs, and in another recently published paper of mine [33] for SDCs and GPBs:

Cimini A, Poliziani A, Morgante L, Moresi M (2024) "Use of malted pulses to formulate gluten-free fresh-egg pasta." Italian Journal of Food Science, 36(1), 105-115. https://doi.org/10.15586/ijfs.v36i1.2451

The composition, antinutrient contents, and in vitro glycemic index are also reported in ref. [33]. A preliminary consumer test was conducted in Rome last October, and it will be repeated on February 21st to gather at least 50 responses, which will be analyzed by experts in the field.

10) Finally, in the conclusions section, the paper discusses the trade-off between reduced cooking times and increased carbon footprint. Can you provide further insights into how this trade-off was evaluated and its implications for consumer choices and environmental sustainability?

At Lines 957-960, the following was reported: However, nutritionally and environmentally conscious consumers should recognize that the energy expended during the malting process, along with the generation of additional processing waste (radicles and cuticles), has led to a 13-17% increase in carbon footprint and Overall Weighted Sustainability Score compared to whole dry legumes. The previous comment - This increase in OWSS was not mitigated by the reduced cooking times of malted and decorticated legumes - was omitted. However, it's worth noting that in Appendix 1, it was indicated that due to the shorter cooking times of malted and decorticated pulses (refer to Table 6), the overall cooking energy decreased to 1.12, 0.98, or 0.85 kWh per kg of malted and decorticated SDCs, GPBs, or OLs, respectively. Additionally, the use of an induction hob would further reduce energy needs to 0.78, 0.70, or 0.62 kWh per kg of malted and decorticated SDCs, GPBs, or OLs, respectively. Such utilization could be bolstered by governmental incentives to be more effective (see also Lines 920-925).

Round 2

Reviewer 1 Report

Comments and Suggestions for Authors

It is better than before, pls proofreading.

Comments on the Quality of English Language

Pls proofreadind